# Disassembly of chiral hydrogen-bonded frameworks into single-unit organometallic helices for enantioselective amyloid inhibition

Yongli Ji[1,6], Caoyu Yang[2,6], Yutong Ye[1], Yin Zhang[3] ✉, Tingting Zhao[1], Shuyue Kong[1], Hongli Chen[1], Pai Liu[1], Zelong Zhao[1], Yilong Li[1], Jing Li[1], Ruixiao Ma[1], Zhiyong Ban[1], Kuo Yuan[4], Zhiyong Tang [2], Yi Liu [1] ✉, Meiting Zhao [5] ✉ & Jun Guo [1] ✉

Chiral nanostructures hold transformative potential across diverse fields, yet their assembly construction remains hindered by the high entropic barrier of dissymmetric building units. Inspired by biological structural dynamics, we construct two chiral copper-based hydrogen-bonded frameworks [*D(L)*·Cu-crystals] via hydrogen-bonded assembly using chiral metal-organic helical as the building unit. Single-crystal X-ray diffraction elucidates hierarchical chirality evolution from asymmetric coordinations to helical chains and framework packing. Furthermore, disassembling *D(L)*·Cu-crystals yields corresponding single-unit chiral metal-organic helices [*D(L)*·Cu-SMOHs], fully exposed active sites and well-preserved helical architectures. Notably, *D(L)*·Cu-SMOHs inhibit amyloid fibrillization effectively with pronounced chirality discrimination, driven by entropy-favored hydrophobic interaction. Molecular docking reveals that *D*-Cu-SMOH exhibits enhanced binding to critical amyloidogenic regions relative to the *L*-enantiomer. This work establishes a dynamic and reversible assembly-disassembly approach applicable for constructions of chiral nanomaterials. Moreover, it provides insights into understanding enantioselective amyloid inhibition, extending applications in asymmetric catalysis, enantioselective separation and chiroptical devices.

Chiral nanostructures, capitalizing on nanoconfinement effects and emergent collective functionalities, are propelling transformative advancements across diverse fields, including asymmetric catalysis[1], enantioselective separation[2], biomedical engineering[3], chiroptical materials[4] and biosensors[5]. While synthetic methodologies for chiral molecules and asymmetric macro-architectures are well established, the rational design and precise construction of chiral structures at the critical nanoscale juncture remain fundamentally challenging[6,7].

[1]State Key Laboratory of Advanced Separation Membrane Materials, School of Chemistry, Tiangong University, Tianjin, P. R. China. [2]CAS Key Laboratory of Nanosystem and Hierarchical Fabrication, CAS Center for Excellence in Nanoscience, National Center for Nanoscience and Technology, Beijing, P. R. China. [3]Department of Chemistry, University of North Texas, 1508 W Mulberry St, Denton, TX, USA. [4]Institute for New Energy Materials and Low Carbon Technologies, Tianjin University of Technology, Tianjin, P. R. China. [5]State Key Laboratory of Advanced Materials for Intelligent Sensing , Tianjin Key Laboratory of Molecular OptoelectronicSciences, Department of Chemistry, Institute of Molecular Aggregation Science, Tianjin University, Tianjin, P. R. China. [6]These authors contributed equally: Yongli Ji, Caoyu Yang. ✉e-mail: yin.zhang@unt.edu.cn; yiliuchem@whu.edu.cn; mtzhao@tju.edu.cn; junguo@tiangong.edu.cn

**Fig. 1 | Assembly and disassembly of chiral hydrogen-bonded frameworks with dynamic structures proposed in this work.** The blue represents *D*-Cu-SMOH, and red represents *L*-Cu-SMOH. The red dashed line indicates hydrogen bonds.

That is the divergent design principles and synthetic strategies across dimensional regimes, distinct from both molecular and bulk chirality. For example, molecular chirality only requires atomic-level manipulation of stereogenic centers/axes/planes[8], whereas macroscopic chirality mainly emerges from the asymmetric alignment of structural motifs[9]. In stark contrast, nanoscale chirality lies on the synergistic orchestration of three interdependent processes, including molecular stereogenesis, hierarchically dissymmetry propagation across dimensional scales and cooperative chirality amplification through collective interactions[10,11]. Self-assembly is well-recognized as a powerful synthetic strategy for constructions of nanostructures with premade compositions, structures, properties and functionalities[12]. Nevertheless, the direct assembly of chiral nanostructures from asymmetric building blocks is still confronting the high entropic barrier stemming from their unsymmetric, non-planar, complex and variable configuration characteristics[13,14]. Unfortunately, typical driving forces during assembly, such as hydrogen bonding, van der Waals and π-π interactions, are insufficient to make desired compensations in thermodynamics[15,16].

Nature provides profound inspiration for the design and assembly of exquisite and versatile chiral nano-hierarchies through dynamical structural transitions and evolutions. For instance, DNA replication initiates from the dynamically de-helixing of prototypical duplex nanostructures to expose single-stranded templates and is subsequently completed by the hydrogen-bonded re-association of nascent DNA strands[17]. Analogously, peptide segments can fold into multi-hierarchical protein nanostructures for function integrations, while well-folded proteins can unfold their assembled structures back into disassembled peptides for facile traversals across membranes[18]. Lessons from chiral dynamics spreading in life, we herein propose an assembly-disassembly strategy to construct chiral framework nanostructures (Fig. 1). First, two types of copper-based frameworks [*D(L)*-Cu-crystals] are assembled adopting a one-dimensional (1D) chiral metal-organic helix as the building unit via hydrogen bonding. The hierarchical chirality evolution of *D(L)*-Cu-crystals is explicitly identified by single-crystal X-ray diffraction (SXRD), revealing the emergence of asymmetric metal coordination, propagation of 1D helix and chiral framework packing. Subsequent disassembly of crystalline *D(L)*-Cu-crystals generates corresponding single-unit chiral metal-organic helices [*D(L)*-Cu-SMOHs]. It is worth emphasizing that conventional X-ray diffraction and electron diffraction techniques face fundamental limitations in resolving subnano and aperiodic structures like disassembled single-unit fibrils[14,19]. Herein, we introduce an integrated electron paramagnetic resonance (EPR) and X-ray absorption fine structure (XAFS) methodology to unambiguously decipher structural parameters of *D(L)*-Cu-SMOHs, including metal coordination environment, electronic configuration, etc. Combined with helicity verification through vibrational circular dichroism (VCD), the well-defined

*D(L)*-Cu-SMOHs hence establish a framework platform for further functional exploration as well as structure-performance correlation studies.

The inhibition and prevention of amyloid fibrillization are regarded as a critical precautionary/therapeutic strategy for neurodegenerative diseases such as Alzheimer's disease and Parkinson's disease[20–22]. Caused by insufficient surface active sites, weak interactions and inexplicable mechanisms, conventional chiral nanomaterials usually resulted in inferior inhibition efficiency and negligible chirality-dependent differentiation on the inhibition of amyloid fibrillization[23,24]. Thanks to their single-unit architectures, fully exposed active sites, as well as well-maintained helical configurations, *D(L)*-Cu-SMOHs were utilized as nanoinhibitors for amyloid fibrilization, exhibiting not only high inhibition efficiency but also significant chirality-discrepant performances. To gain in-depth mechanistic insights, nanowatt isothermal titration calorimetry (ITC) and temperature-dependent fluorescence spectroscopy (FL) were employed to study the correlated thermodynamic interactions between *D(L)*-Cu-SMOHs and amyloid. The results revealed an entropy-favored interaction mechanism ($\Delta H > 0$ and $\Delta S > 0$) driven by enantioselective hydrophobic π-π stacking, with *D*-Cu-SMOH exhibiting a more favored Gibbs energy change compared to its *L*-enantiomer. Furthermore, molecule docking simulation has further exquisitely depicted the significantly enantioselective π-π stacking between the aromatic pyridine group of *D(L)*-Cu-SMOHs and hydrophobic amino acid residues of amyloid, which are of essential amyloidogenesis regions. In other words, this work advances beyond constructing hierarchical chiral framework nanostructures via a dynamic and reversible assembly-disassembly protocol. More critically, leveraging their well-defined structures, adaptive helical conformations, and thermodynamic interaction mechanisms, the disassembled single-unit chiral nanomaterials are anticipated to extend potent and chirality-dependent functionalities and performances unavailable from assembled counterparts.

## Results
### Assembly and disassembly of chiral HOFs
Two types of enantiomeric copper-based hydrogen-bonded frameworks, i.e., *D(L)*-Cu-crystals, were assembled inspired by the hierarchical chiral structures of proteins. In detail, enantiomeric *N*-(4-pyridylmethyl)-threonine, abbreviated as *D(L)*-Py-Thr, were synthesized through the Schiff-base reaction between *D(L)*-threonine and 4-pyridinecarboxaldehyde following a further imine reduction by NaBH₄ (for details see Methods; Supplementary Figs. 1–7, Supplementary Table 1). Subsequently, rectangular blue crystals of *D(L)*-Cu-crystals with bulk dimensions both up to 150 μm in both length and width and 50 μm in height (Supplementary Fig. 8) were cultivated in a mixed methanol-water solution containing *D(L)*-Py-Thr

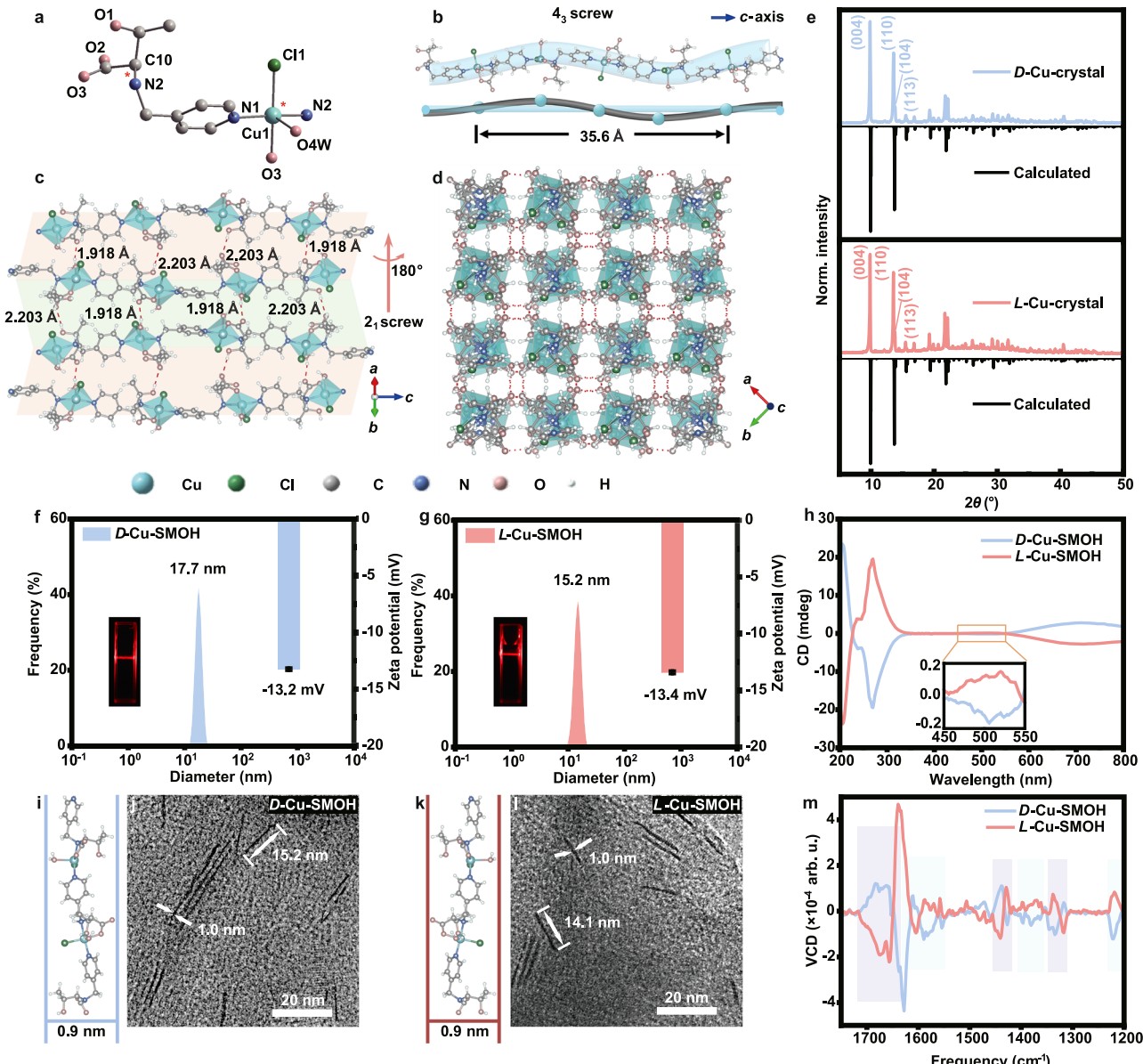

**Fig. 2 | Dynamic evolution of assembly and disassembly of *D(L)*-Cu-crystals.**
**a** Asymmetric coordination mode of *D*-Cu-crystal with red asterisks indicating chiral centers. **b** The 1D helical unit featuring a left-handed $4_3$ screw with a pitch of 35.6 Å along the *c*-axis direction. **c** 2D lamella structure assembled by interchain hydrogen bonding. The shaded area indicates the $2_1$ screw. **d** 3D framework structure projected in the *a*−*b* plane. Red dashed lines in (**c**) and (**d**) indicate hydrogen bonds. Element color: C is gray; O is pink; N is deep blue; H is white; Cl is green; Cu is light blue. **e** PXRD patterns of *D(L)*-Cu-crystals alongside calculated patterns. **f**, **g** DLS characterization of *D(L)*-Cu-SMOHs dispersed in water and the inset shows the related Tyndall effect. **h** CD spectra of *D(L)*-Cu-SMOHs dispersed in water. **i** Corresponding crystallographic dimensions of *D*-Cu-SMOH single-unit helix according to SXRD. **j** HR-TEM image of *D*-Cu-SMOH dispersed in water. **k** Corresponding crystallographic dimensions of *L*-Cu-SMOH single-unit helix according to SXRD. **l** HR-TEM image of *L*-Cu-SMOH dispersed in $H_2O$. **m** VCD spectra of *D(L)*-Cu-SMOHs. The purple shaded region represents signals from the carboxylate group, while the blue shaded region represents signals from the Py ring. Source data are provided as a Source Data file.

and $Cu(CH_3COO)_2$ at 60 °C (see Methods). In good line with their rectangular crystal morphologies, SXRD analyses demonstrate that both *D*-Cu-crystal (CCDC 2330456) and *L*-Cu-crystal (CCDC 2330457) belong to the tetragonal crystallographic phase with the chiral space group of $P4_32_12_1$ and $P4_12_12_1$, respectively (Supplementary Tables 2–13). Considering the identical coordination fashions, we therefore only discuss the structural evolution of the representative *D*-Cu-crystal in the following paragraph (discussion of *L*-conformer is available in Supplementary Fig. 9).

Distinct from the typical planar square coordination among reported Cu-based frameworks[25,26], the Cu(II) center in *D*-Cu-crystal adopts a distorted trigonal-bipyramidal (TBP) coordination by one

pyridine nitrogen (N1) from the first *D*-Py-Thr ligand, one amino nitrogen (N2), one carboxylate oxygen (O3) from the second *D*-Py-Thr ligand, one water oxygen (O4W) and an additional chlorine (Cl1) ion for final charge balance (Fig. 2a). By means of the extended coordination of *D*-Py-Thr, the formed asymmetric unit $[Cu(D\text{-}Py\text{-}Thr)(Cl)(H_2O)]_∞$ is evolved into a one-dimensional (1D) infinite metal-organic helix featuring a left-handed fourfold ($4_3$) screw with a pitch of 35.6 Å along the crystallographic *c*-axis (Fig. 2b). Interestingly, the associated $[Cu(D\text{-}Py\text{-}Thr)(Cl)(H_2O)]_∞$ unit is highly analogous to the well-known α-helix assembled via peptide condensation. Similar to the α-helix assembling into β-sheet via hydrogen bonding, the adjacent 1D helix further forms the two-dimensional (2D) lamella through two types of hydrogen

bonding in a complementary relationship (Fig. 2c). That is the hydrogen bonding between the hydroxyl hydrogen and the uncoordinated acetate oxygen (O1-H1---O2-C11, 2.203 Å) and the hydrogen bonding between the coordinated water hydrogen and the uncoordinated acetate oxygen (O4w-H4A---O2-C11, 1.918 Å). Being subject to a twofold screw ($2_1$) symmetry in crystallography, above helix pairs within 2D lamellar are arranged in the right-handed $2_1$ chiral configuration along the crystallographic $a$-axis (shaded helices in Fig.1c). On account of the tetragonal phase, such hydrogen bonding appears in a perpendicular and intersecting fashion within the crystallographic $a$-$b$ plane and therefore constructs the final three-dimensional (3D) hydrogen-bonded framework using 1D [Cu($D$-Py-Thr)(Cl)(H$_2$O)]$_\infty$ helix as the building unit (Fig. 2d).

The scale-up production of $D(L)$-Cu-crystal powder proceeded via simply increasing precursor concentrations (see Methods). $D(L)$-Cu-crystal powder with dimensions of ca. 16 μm in both length and width and ca. 4 μm in height are synthesized (Supplementary Figs. 10, 11). According to Powder X-ray diffraction (PXRD) characterizations, nearly identical diffraction patterns are observed for $D(L)$-Cu-crystal powder owing to their identical crystallography, albeit opposite chiral space group. The observed patterns are also well consistent with the calculated ones derived from SXRD results, presenting four main peaks centered at $2\theta$ values of 9.94°, 13.72°, 13.90°, and 15.63° corresponding to the diffraction of (004), (110), (104) and (113) facet, respectively (Fig. 2e). In addition, nearly identical Fourier transform infrared (FTIR) spectra (Supplementary Fig. 12) and thermogravimetric analysis (TGA) curves (Supplementary Figs. 13, 14) are acquired for $D(L)$-Cu-crystal enantiomers, further confirming their identical coordination environments and thermal stabilities. The chiral characteristics of $D(L)$-Cu-crystals were investigated using circular dichroism (CD) spectroscopy. As expected, $D(L)$-Cu-crystals exhibit strong and mirror-symmetric CD activities (Supplementary Fig. 15) in the wavelength range of 210 to 790 nm, a typical indicator of enantiomeric relationship with each other. In specification, the CD peak centered at 220 nm is attributed to the $n \rightarrow \pi^*$ transition of the carboxylate group[27]. While CD signals around 270 nm are attributed to the $\pi \rightarrow \pi^*$ and $n \rightarrow \pi^*$ electron transitions of the pyridyl ring, which are involved in a homohelical configuration[27]. In sharp contrast, weak chiroptical responses are observed for raw $D(L)$-Py-Thr ligands (Supplementary Fig. 15) at the optical window of pyridyl groups due to the lack of such helical configuration. On account of the asymmetric coordination mode of central Cu(II), CD signals between 300–350 nm arise from the ligand-to-metal charge transfer (LMCT)[28]. While CD responses appearing in visible regions arise from $d$–$d$ transitions of the asymmetric Cu(II) center featuring an incompletely occupied $3d^9$ configuration[29].

To expose abundant chiral active sites and facilitate intimate interactions, we propose the disassembly of $D(L)$-Cu-crystals into single-unit metal-organic helices [$D(L)$-Cu-SMOHs] via mimicking the unfolding process of proteins into peptide segments. The disassembly of $D(L)$-Cu-crystals readily proceeded by adopting water as the protic solvent for competitively breaking the interchain hydrogen bonding among [Cu($D$-Py-Thr)(Cl)(H$_2$O)]$_\infty$ helix units (see Methods). The resulting $D(L)$-Cu-SMOHs show a pronounced Tyndall effect in water under red laser irradiation (inset shown in Fig. 2f, g). Accordingly, the dynamic light scattering (DLS) measurements for $D(L)$-Cu-SMOHs demonstrate related hydrodynamic sizes of 17.7 nm and 15.2 nm, respectively (Fig. 2f, g). The zeta potentials of $D(L)$-Cu-SMOHs (ca. −13 mV) also claim their dispersion stability (Fig. 2f, g). Moreover, high-resolution transmission electron microscopy (HR-TEM) was performed on $D(L)$-Cu-SMOHs to directly visualize their morphologies and sizes. Absolutely distinct from pristine $D(L)$-Cu-crystals featuring bulk rectangular shapes, $D(L)$-Cu-SMOHs (Fig. 2j, l) display 1D morphologies with chain lengths of 15.2 nm and 14.1 nm, respectively, which are in good line with the obtained DLS sizes. Significantly, the

diameters of both $D(L)$-Cu-SMOHs are measured to be about 1.0 nm, in good agreement with the crystallographic width of a single unit [Cu($D(L)$-Py-Thr)(Cl)(H$_2$O)]$_\infty$ (0.9 nm, Fig. 2i, k). Strong CD signals (Fig. 2h) shown by $D(L)$-Cu-SMOHs imply their well-maintained chirality. The corresponding asymmetric $g$-factors of $D(L)$-Cu-SMOHs are calculated to be $3.7 \times 10^{-3}$ and $1.2 \times 10^{-2}$ centered at 302 nm and 694 nm, respectively (Supplementary Fig. 16). Similar structural features (Supplementary Figs. 17–19) are also preserved for $D(L)$-Cu-SMOHs under simulated physiological conditions (i.e., 10 mM phosphate buffer, pH = 7.4 and 37 °C). Notably, the CD response of $D(L)$-Cu-SMOHs also significantly differs from those of raw $D(L)$-Py-Thr ligands as well as simple coordination complexes formed between Cu(II) and $D(L)$-Py-Thr ligand under the same precursor concentrations (Supplementary Fig. 20). Moreover, the specific rotation $[\alpha]_D^{25}$ of $D(L)$-Cu-SMOHs are measured to be +51.2° and −50.4°, respectively, which are also obviously larger than +24.5°(−25.3°) of $D(L)$-Py-Thr ligands and +35.2°(−36.1°) of the simple coordination complex, affirming the collective chiroptical activities of $D(L)$-Cu-SMOHs (Supplementary Table 1). We further employed VCD spectroscopy to resolve whether helical conformation is preserved by $D(L)$-Cu-SMOHs[30,31]. Remarkably, mirror-symmetric VCD signals (Fig. 2m), especially those corresponding to skeleton vibrations of the planar and axial pyridine ring (Supplementary Fig. 21 and Supplementary Table 14), definitely claim its involvement in intrinsic helical conformations.

## Structure characterizations of disassembled SMOHs

EPR spectroscopy is a powerful tool to accurately characterize electronic structures and coordination environments, particularly for transition-metal species featuring unpaired electrons[32]. Hence, the EPR spectrum of the disassembled $D(L)$-Cu-SMOHs was measured and compared with that of pristine $D(L)$-Cu-crystals. As for $D$-Cu-crystals prior to disassembly, the obtained EPR spectra exhibit a typical axial symmetry with two anisotropic Landé $g$-factors (Fig. 3a). That is $g_\parallel$ factor with the complex's $z$-axis parallel to the applied magnetic field and $g_\perp$ with the complex's $z$-axis perpendicular to the applied magnetic field. According to the well-fitted simulation spectrum, the assigned $g_\perp$ (2.17) is obviously larger than $g_\parallel$ (2.05) which is nearly coincident with $g_e$ (i.e., the $g$-factor of free electron). In combination with the $3d^9$ electronic configuration of Cu(II) as well as the crystal field theory[33] (see Methods, Supplementary Fig. 22, and Supplementary Table 15), the observed EPR characterized by a $g_\perp > g_\parallel \approx g_e$ relation demonstrates a spin electron-occupied $d_{z^2}$ ground state and a triangular bipyramid (TBP) coordination[34] of $D$-Cu-crystal (Fig. 3c), in good line with the 5-coordination mode resolved by SXRD. Interestingly, the EPR spectrum of disassembled $D$-Cu-SMOH shows two principal components featuring distinct coordination environments according to the fitted EPR spectra (Fig. 3b). First, disassembled $D$-Cu-SMOH retains the original TBP coordination configuration that is coincident with $D$-Cu-crystal. Besides, a 6-coordinated elongated octahedron (EO, the model is shown in Supplementary Fig. 23) EPR characteristic of $g_\parallel = 2.23 > g_\perp = 2.09 > g_e$ and a spin electron-occupied $d_{x^2-y^2}$ as the ground state is additionally discerned for $D$-Cu-SMOH (Fig. 3d), indicating a transformation from TBP to EO fashion involved by H$_2$O coordination during the disassembly process[35]. Note that the estimated $d_{yz}/d_{xz} - d_{x^2-y^2}$ transition gap (2.3 eV) of EO Cu(II) shows good agreement with the observed visible $d$-$d$ absorption band (2.2 eV) of $D$-Cu-SMOH in Supplementary Fig. 24.

To validate the EPR findings, we further acquired XAFS spectroscopy data for both $D$-Cu-crystal and $D$-Cu-SMOH samples to enable precise structure characterization and comparison. The Cu $K$-edge X-ray absorption near-edge structure (XANES) spectra of both $D$-Cu-SMOH and $D$-Cu-crystal show nearly identical absorption patterns to that of CuO standard sample, verifying the same Cu(II) oxidation states among them (Fig. 3e). Close inspection finds a weak pre-edge ($1s \rightarrow 3d$) quadrupolar transition[36] around 8977 eV for both $D$-Cu-crystal and $D$-

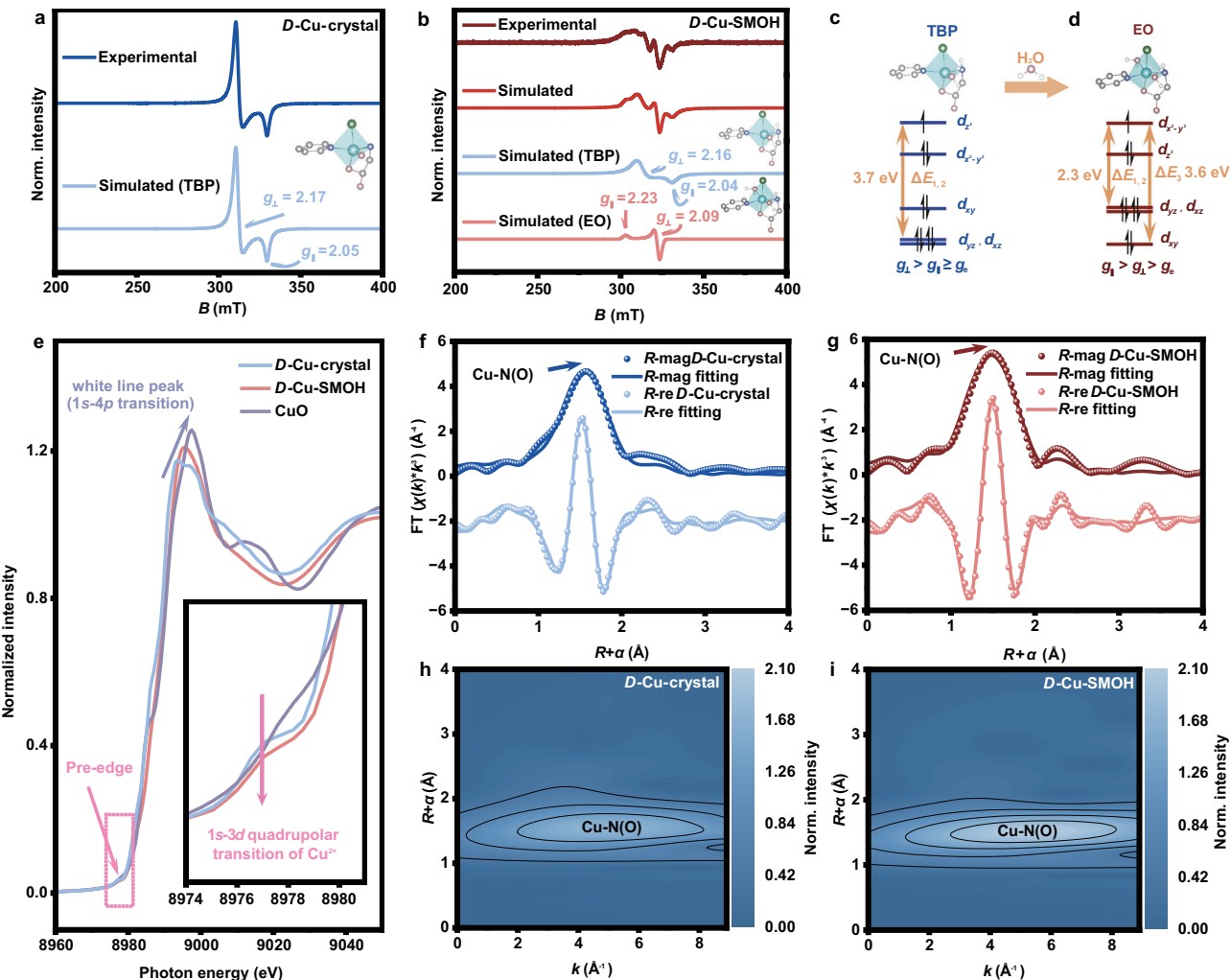

**Fig. 3 | Structure Characterizations of *D(L)*-Cu-SMOHs. a** The EPR spectra of *D*-Cu-crystal. **b** The EPR spectra of *D*-Cu-SMOH. **c** Electronic orbital aligns of Cu(II) in the TBP crystal filed. "$\Delta E_{1,2} = 3.7$ eV" is the energy gap between the ground-state $d_{z^2}$ and the $d_{xz}$ or $d_{yz}$ orbitals. **d** Electronic orbital aligns of Cu(II) in the *E*O crystal filed. "$\Delta E_{1,2} = 3.6$ eV" is the energy gap between the ground-state $d_{x^2-y^2}$ and the $d_{xz}$ or $d_{yz}$ orbitals, while "$\Delta E_3 = 2.3$ eV" is the energy gap between the ground-state $d_{x^2-y^2}$ and the $d_{xy}$ orbital. **e** Cu *K*-edge XANES spectra for *D*-Cu-crystal and *D*-Cu-SMOH. The inset shows the Pre-edge peak. The white line peak is $1s \rightarrow 4p$ dipolar transition. **f** Corresponding Fourier transformed EXAFS spectra with fitting lines in non-phase corrected *R* space for *D*-Cu-crystal. **g** Corresponding Fourier transformed EXAFS spectra with fitting lines in non-phase corrected *R* space for *D*-Cu-SMOH. **h** Corresponding wavelet transform map of EXAFS spectra for *D*-Cu-crystal. **i** Corresponding wavelet transform map of EXAFS spectra for *D*-Cu-SMOH. Source data are provided as a Source Data file.

Cu-SMOH while the absorption intensity of *D*-Cu-SMOH is obviously weakened in comparison to that of *D*-Cu-crystal (inset in Fig. 3e). A stronger pre-edge absorption means a larger structural deviation from centro-symmetry due to the forbidden $1s \rightarrow 3d$ dipolar transition according to the Laporte selection rule[37,38], which is consistent with the coordination transition from a low centrosymmetric TBP to a high centrosymmetric EO occurring in the $H_2O$ triggered disassembly process. Meanwhile, the intensity enhancement of white line peaks ($1s \rightarrow 4p$ dipolar transition) of *D*-Cu-SMOH relative to that of *D*-Cu-crystal further evinces an increase in the coordination number of the former after disassembly[39–41]. Subsequently, we investigated their extended X-ray absorption fine structures (EXAFS) to further elucidate fine changes in the local coordination geometry. Notably, oscillation patterns with minimal disparities are obtained for *D*-Cu-SMOH and *D*-Cu-crystal in the *k*-space, revealing that disassembled *D*-Cu-SMOH well inherits the coordination structures of parent *D*-Cu-crystal (Supplementary Fig. 25). However, the corresponding coordination number of *D*-Cu-SMOH becomes larger than that of *D*-Cu-crystal (Fig. 3f, g) according to the quantitative fitting results (Supplementary Table 16). In detail, the *R*-space peak centered at 1.5 Å assigned to Cu-N(O)

bonding is fitted according to the SXRD-resolved crystal structure and is derived to a coordination number of ca. 3.7 for *D*-Cu crystal. Remarkably, associated Cu-N(O) bonding appearing at 1.5 Å is also fitted for disassembled *D*-Cu-SMOH but with a much larger coordination number of 4.6, unambiguously claiming additional $H_2O$ coordination after disassembly. A consistent result can also be attained from the wavelet transform (WT) results. One can directly visualize that the WT maxima of *D*-Cu-SMOH correlated to the Cu-N(O) bonding are significantly higher than those of the raw *D*-Cu-crystal (Fig. 3h, i). Taking the EPR and XAFS results together, we can conclude that the *D*-Cu-crystal, upon disassembly, transforms into the corresponding 1D metal-organic helix i.e., $[Cu(D\text{-Py-Thr})(Cl)(H_2O)]_\infty$ building unit accompanied by a portion of Cu(II) additionally coordinated by exoteric $H_2O$ molecules. Considering the dynamic feature of hydrogen bonding, we further explored whether disassembling reversibility occurs in *D(L)*-Cu-SMOHs. Very interestingly, introducing aprotic solvent (e.g., acetonitrile) triggers the water desolvation from *D(L)*-Cu-SMOHs, promoting interchain hydrogen bonding and reformation of blue precipitates (see Methods). PXRD (Supplementary Fig. 26) and SEM (Supplementary Fig. 27) characterizations of the centrifugated

precipitates confirm reassembled crystalline structures identical to parent *D(L)*-Cu-crystals, unequivocally demonstrating reversible assembly-disassembly cycling between *D(L)*-Cu-crystals and *D(L)*-Cu-SMOHs (Fig. 1).

## Enantioselective inhibition of amyloid fibrillization

The inhibition of the amyloid fibrillization process is considered an efficient route to alleviate and even treat neurodegenerative diseases. As a representative amyloid protein, human insulin (HI) is characterized by a well-defined amino acid sequence and 3D configurational structure and is therefore widely employed as the model to study the fibrillization process of amyloid-like proteins[42]. The inhibition effect of *D(L)*-Cu-SMOHs on HI fibrillization was performed by using a classic thioflavin T (ThT) fluorescence probe, which increases its fluorescence intensity upon binding to mature amyloid fibrils[43]. As the background, *D(L)*-Cu-SMOHs themselves indeed do not show additional fluorescence under the employed experimental conditions (Supplementary Fig. 28). Then, the dependences of *D(L)*-Cu-SMOHs against HI fibrillization were monitored in the concentration range of 0-200 μg mL$^{-1}$ (Fig. 4a) which demonstrates a pronounced enantioselective effect at the concentration of 50 μg mL$^{-1}$. Meanwhile, the 3-(4,5-dimethylthiazol-2-yl)−2,5-diphenyltetrazolium bromide (MTT) assay (Fig. 4b) was utilized to demonstrate the low toxicity of *D(L)*-Cu-SMOHs on SH-SY5Y cells under such concentration ranges[44]. According to the time-monitored ThT curves (Fig. 4c), free HI greatly increases its FL intensity during the incubation period, accompanied by the massive production of amyloid fibrils. Significantly, HI incubated with *L*-Cu-SMOH and *D*-Cu-SMOH both show obviously lower FL intensity with a normalized FL ratio of 40% and 30%, respectively (Supplementary Table 17). The more pronounced inhibitory effect acquired by *D*-Cu-SMOH than *L*-Cu-SMOH is further demonstrated by the fluorescence stopped-flow kinetic studies (Supplementary Fig. 29). In which, one can see that a faster FL decay curve is observed for *D*-Cu-SMOH (21.70 s$^{-1}$), confirming its faster binding rate to HI than that of the *L*-Cu-SMOH (8.56 s$^{-1}$).

It is well accepted that protein fibrillization is characterized by a proportional decrease in α-helix content and a significant proportional increase in β-sheet content. Hence, CD spectroscopy was employed to precisely observe the secondary structure evolution of HI during fibrillization[45]. The pristine HI monomer exhibits negative CD peaks around 208 and 222 nm, which are assigned to the two characteristic signs of α-helix (Fig. 4d)[46]. As a pre-incubation control, the unaltered CD spectra of HI (Supplementary Fig. 30) acquired immediately after addition of *D(L)*-Cu-SMOHs exclude intrinsic CD influences arising from themselves at the employed concentration. The fibrillization of HI after incubation involves significant transformation of β-sheet from α-helix, which is clearly monitored by the appearance of a broad band near 219 nm (Fig. 4d)[47]. In cases of HI samples incubated with *D(L)*-Cu-SMOHs, the two negative α-helix signs of HI are still retained, free from appearing typical β-sheet CD sign around 219 nm. Apparently, the retained CD intensity by *D*-Cu-SMOH is also stronger than that of *L*-Cu-SMOH, further demonstrating the higher enantioselective inhibition effect of *D*-Cu-SMOH on HI fibrillization. We further quantitatively calculate the proportions of secondary structures during HI fibrillization according to statistical methods. On average, the constituent of β-sheet increased from 25.5% ± 1.2% in HI monomers to 39.1% ± 0.2% after HI fibrillization, while the α-helix proportion decreased from 27.7% ± 0.1% to 9.0% ± 0.1%. Remarkably, the corresponding proportion of β-sheet increased from 25.5% ± 1.2% to only 26.8% ± 1.2% particularly in the coincubation sample of *D*-Cu-SMOH (Fig. 4e). Consistently, the α-helix proportion was also not obviously declined (26.4% ± 0.2%) for *D*-Cu-SMOH in comparison to the HI monomer. The FT-IR band associated with the amide region is critical for analyzing the evolution of secondary structures of HI as well[48]. Prior to fibrillization, the HI monomer exhibits a characteristic peak at 1657 cm$^{-1}$ corresponding to the amide band of α-helix (Fig. 4f)[49]. After incubation of HI alone, a

bathochromic shift of the C = O vibration from 1660 cm$^{-1}$ to the lower wavenumber band around 1630 cm$^{-1}$ can be observed, attributable to the formation of β-sheets in HI fibrils via hydrogen bonding[50,51]. There are no obvious vibration peaks around 1630 cm$^{-1}$ under introductions of *D(L)*-Cu-SMOHs, particularly for *D*-Cu-SMOH, reaffirming a stronger inhibition effect. Consistent enantioselective inhibitory potency can be more remarkably discerned using VCD spectroscopy (Supplementary Fig. 31). Besides, the enantioselective inhibition effect of *L*-Cu-SMOH and *D*-Cu-SMOH on HI fibrillization is further visualized using microscopy imaging technologies. In line with the ThT results, confocal laser scanning microscopy (CLSM) images (Supplementary Fig. 32) intuitively present the inhibition effect on HI fibrilization upon introduction of disassembled frameworks with a more prominent FL decrease by *D*-Cu-SMOH than that of *L*-Cu-SMOH. Moreover, transmission electron microscopy (TEM) and atomic force microscopy (AFM) are utilized to inspect the morphology change of HI during fibrillization. As shown in Fig. 4g, mature protofibrils of ca. 3.51 μm in length are discerned in the sole HI sample after incubation, which is consistent with its large DLS size of 3.75 μm (Supplementary Fig. 33). When HI is incubated with either *D*-Cu-SMOH or *L*-Cu-SMOH, the amount of HI protofibrils diminishes significantly due to their significant inhibition against fibril formations. More notably, only spherical HI aggregates can be discerned in the case of *D*-Cu-SMOH, agreeing well with its more pronounced inhibition effect. Accordingly, the relevant DLS size of HI was only 0.35 μm and 0.65 μm in the case of co-incubation with *D*-Cu-SMOH and *L*-Cu-SMOH, respectively. Consistent conclusions can also be drawn based on AFM images (Fig. 4h). Micrometer-sized amyloidogenic fibrils can be seen in the matured HI sample. Upon coincubation, only a few globular dots are observed, particularly in the case of *D*-Cu-SMOH. Control experiments using free chiral ligands (*D(L)*-Py-Thr) and an achiral Cu-based framework (Cu-Gly-MOF, Supplementary Fig. 34)[52] further highlight the intrinsic anti-amyloid activity of *D(L)*-Cu-SMOHs. Complementary characterizations involving ThT assay (Supplementary Fig. 35), CD spectroscopy (Supplementary Fig. 36), TEM morphology (Supplementary Fig. 37) and DLS analysis (Supplementary Fig. 38) consistently show negligible or significantly reduced inhibitions of HI fibrillization by controls. These convergent results unambiguously establish the prominent enantioselective inhibition exhibited by *D(L)*-Cu-SMOHs.

## Thermodynamic study between amyloid and SMOHs

As the golden criterion for thermodynamic study, ITC is competent to quantitatively measure thermodynamics involved in complex interactions, including electrostatic interaction, hydrogen bonding, van der Waals forces, and hydrophobic forces[53]. Hence, nanowatt ITC was carefully used to measure the thermodynamic parameters during the affinities of disassembled frameworks to HI monomers. Notably, titration of HI monomers with either *D*-Cu-SMOH (Fig. 5a) or *L*-Cu-SMOH (Fig. 5b) consistently affords positive thermograms, indicative of endothermic interaction processes. Their integrated enthalpy curves are further fitted by a typical single-site model, which consistently gives positive enthalpy changes ($\Delta H > 0$) and large positive entropy changes ($T\Delta S \gg 0$)[54], but final negative Gibbs free energy changes ($\Delta G < 0$) as the compensation (Table 1). Totally distinct from the enthalpically favored interactions (e.g., electrostatic force or/and hydrogen bonding)[55], our ITC results ($\Delta H > 0$ and $\Delta S \gg 0$) suggest an entropy-driven interaction (i.e., hydrophobic affinity stemming from desolvation effect) between HI and disassembled framework according to Ross and Subramanian's theory[56]. To eliminate alternative entropy gains, a series of control ITC experiments (Supplementary Figs. 39–44) involving sole chiral *D(L)*-Py-Thr ligands ($\Delta H < 0$ and $\Delta S > 0$), achiral pyridyl-containing ligand (Py -Gly, Supplementary Figs. 45–48) ($\Delta H < 0$ and $\Delta S > 0$) and additional Cu-framework (Cu-Gly-MOF, Supplementary Fig. 34) free of both chirality and aromaticity ($\Delta H < 0$ and $\Delta S < 0$) have been further conducted. According to results

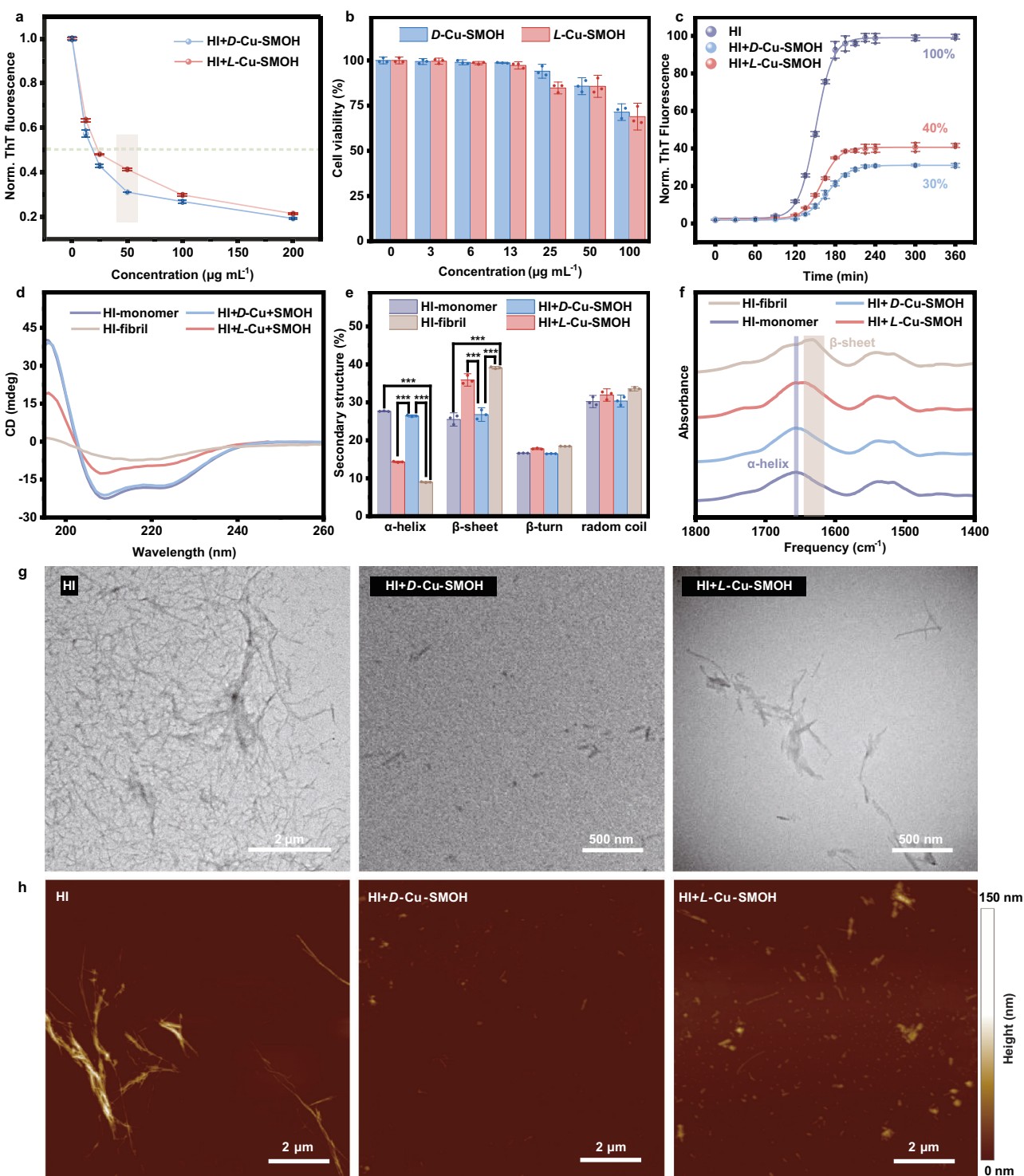

**Fig. 4 | Enantioselective inhibition of *D(L)*-Cu-SMOHs on HI fibrillization.**
**a** Concentration-dependent inhibition effect of *D(L)*-Cu-SMOHs on HI fibrillization.
**b** MTT cytotoxicity of *D(L)*-Cu-SMOHs. **c** Kinetics of HI Fibrillization monitored by
the ThT assay in the absence/presence of *D(L)*-Cu-SMOHs. **d** CD spectra of HI
without or with *D(L)*-Cu-SMOHs incubation. **e** Secondary structure analysis from CD
spectrum. **f** FT-IR spectra of HI without or with *D(L)*-Cu-SMOHs incubation. **g** TEM
imaging of HI without and with *D(L)*-Cu-SMOHs incubation. **h** AFM imaging of HI
without and with *D(L)*-Cu-SMOHs incubation. Error bars indicate the s.d. (*n* = 3
independent samples). *$P < 0.05$, **$P < 0.01$, ***$P < 0.001$, two-sided Student's *t* test.
Source data are provided as a Source Data file.

summarized in Supplementary Table 18, all controls accordingly present exothermic thermograms ($\Delta H < 0$) in stark contrast to the endothermic signature ($\Delta H > 0$) of *D(L)*-Cu-SMOHs, thereby eliminating enthalpically driven interactions (e.g., electrostatic interactions and hydrogen bonding) between *D(L)*-Cu-SMOHs and HI. Furthermore, entropy gains ($\Delta S > 0$) persist in Py-containing ligands, including *D(L)*-

Py-Thr and Py-Gly while a reversed entropy decrease ($\Delta S < 0$) is observed for Cu-Gly-MOF, without Py group, highlighting the Py-mediated hydrophobic interactions as the critical entropy source.

Furthermore, the affinity constants ($K_a$) are measured to be $1.58 \times 10^5 \, M^{-1}$ and $0.47 \times 10^5 \, M^{-1}$, corresponding to $\Delta G = -29.67 \, \text{kJ mol}^{-1}$ and $\Delta G = -26.65 \, \text{kJ mol}^{-1}$ for *D*-Cu-SMOH and *L*-Cu-SMOH, respectively,

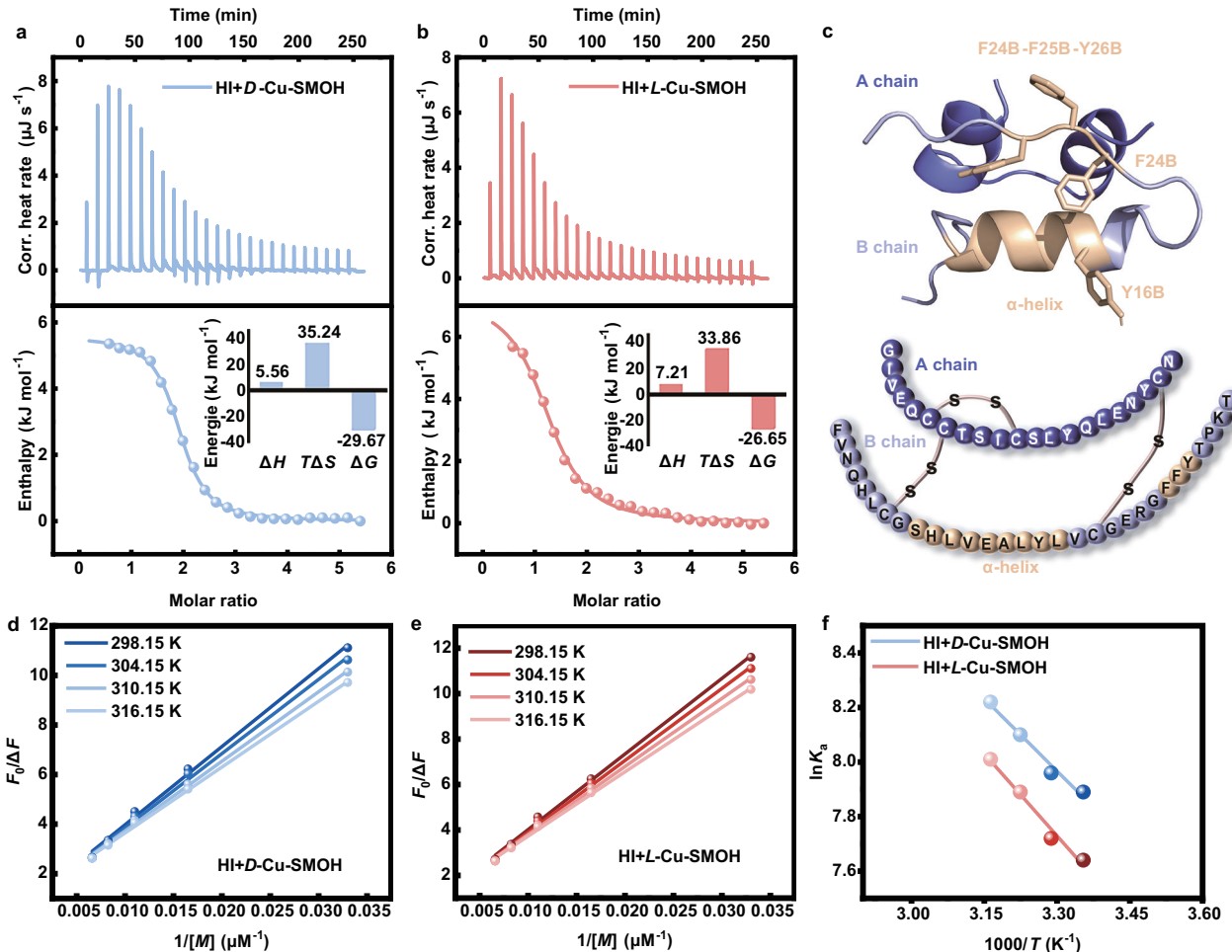

**Fig. 5 | Thermodynamic interactions between $D(L)$-Cu-SMOHs and HI. a** ITC thermograms in the titration of $D$-Cu-SMOH into HI solution at 298.15 K. **b** ITC thermograms in the titration of $L$-Cu-SMOH into HI solution at 298.15 K. Fitting curves are obtained using the unit-point binding model. **c** The 3D structure and amino acid sequence of HI monomer. The key hydrophobic regions (α-helix and FFY regions) are highlighted by orange color. **d** The modified Stern-Volvker fitting

FL quenching plot at varying temperatures for incubation of HI with $D$-Cu-SMOH. **e** The modified Stern-Volmer fitting FL quenching plot at varying temperatures for incubation of HI with $L$-Cu-SMOH. **f** Van't Hoff plot fitting the thermodynamic interaction of HI with $D$-Cu-SMOH ($\ln K_a = \frac{114.82 \text{Jmol}^{-1}\text{K}^{-1}}{R} - \frac{14.73 \text{kJmol}^{-1}}{RT}$) or $L$-Cu-SMOH ($\ln K_a = \frac{119.39 \text{Jmol}^{-1}\text{K}^{-1}}{R} - \frac{16.69 \text{kJmol}^{-1}}{RT}$). Source data are provided as a Source Data file.

**Table 1 | Interactions thermodynamics between HI and $D(L)$-Cu-SMOHs at 298.15 K**

| Method | Sample | $K_a$ (M⁻¹) | ΔH (kJ mol⁻¹) | TΔS (kJ mol⁻¹) | ΔG (kJ mol⁻¹) |
|---|---|---|---|---|---|
| ITC | HI + $D$-Cu-SMOH | 1.58×10⁵ | 5.56 | 35.24 | −29.67 |
| | HI + $L$-Cu-SMOH | 4.67×10⁴ | 7.21 | 33.86 | −26.65 |
| FL | HI + $D$-Cu-SMOH | 2.61×10³ | 14.73 | 34.24 | −19.51 |
| | HI + $L$-Cu-SMOH | 2.07×10³ | 16.69 | 35.58 | −18.89 |

also highlighting a stronger affinity of $D$-Cu-SMOH to HI. According to its well-defined 3D structure as well as amino acid sequence[57,58], one can figure out abundant hydrophobic residues (orange-highlighted regions in Fig. 5c) such as S9B (Ser) to Y16B (Tyr) residues from the α-helix region and typical FFY region namely the F24B (Phe), F25B (Phe), and Y26B (Tyr) residues in the B chain of HI[59–61]. Noteworthily, α-helix structure is the essential region required for transformation into β-sheet secondary structure during the fibrillization process[62]. Equally importantly, FFY tripeptide is also an established aggregation region to accelerate HI fibrillization on account of its strong intermolecular π-π interactions[63,64]. As a result, the aforementioned hydrophobic sites are amyloidogenic-essential regions due to their self-assembly-triggered fibrillizations[59,62,65].

Considering the fluorescence of HI primarily originates from chromophores like Tyr and Phe residues, which can readily interact with aromatic groups of $D(L)$-Cu-SMOHs through hydrophobic interactions and π-π stacking[66]. Hence, the FL quenching protocol[67] is very reliable for probing possible hydrophobic interactions between HI and $D(L)$-Cu-SMOHs. Supplementary Fig. 50a, b presents quenched FL spectra at 298.15 K with peaks around 302 nm upon the addition of $D$-Cu-SMOH and $L$-Cu-SMOH, respectively. The blank FL signals of $D(L)$-Cu-SMOHs (Supplementary Fig. 49) signify negligible background influences. As a result, a modified Stern-Volmer equation is accepted for acquisitions of the corresponding apparent binding constant $K_a$ as the linearity derivation at the high concentration range (Supplementary Fig. 50c, d), also an indication of a mixed quenching

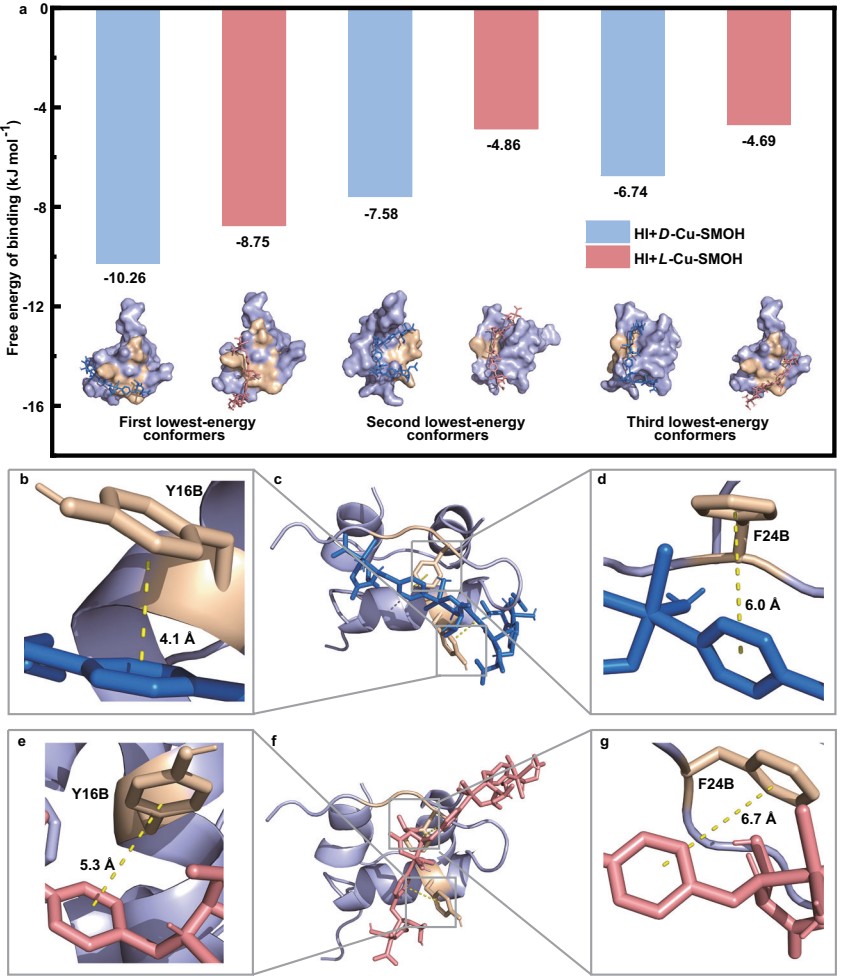

**Fig. 6 | Molecular docking results between HI with *D(L)*-Cu-SMOHs. a** The corresponding three lowest energy levels of docking *D(L)*-Cu-SMOHs onto HI. **b** Locally amplified image of the interaction of Y16B in HI with *D*-Cu-SMOH. **c** The lowest energy conformation of docking *D*-Cu-SMOH onto HI. **d** Locally amplified image of the interaction of F24B in HI with *D*-Cu-SMOH. **e** Locally amplified image of the interaction of Y16B in HI with *L*-Cu-SMOH. **f** The lowest energy conformation of docking *L*-Cu-SMOH onto HI. **g** Locally amplified image of the interaction of F24B in HI with *L*-Cu-SMOH. The yellow dashed line indicates π-π stacking. Source data are provided as a Source Data file.

mechanism[68]. According to the fitted $K_a$ and related $\Delta G$ at varying temperatures (Fig. 5d,e and Supplementary Figs. 50–53), we further calculate the $\Delta H$ and $\Delta S$ thermodynamic parameters using the well-known van't Hoff equation (Fig. 5f and Table 1). The consistent $\Delta H > 0$ and $\Delta S > 0$ results in both *D*-Cu-SMOH and *L*-Cu-SMOH further confirm their pivotal hydrophobic interactions with HI. Moreover, the more negative $\Delta G$ values acquired by *D*-Cu-SMOH over *L*-Cu-SMOH at all measured temperatures also declare a higher affinity for *D*-Cu-SMOH to HI (Table 1 and Supplementary Table 19).

**Theoretical understanding of enantioselective interaction**
To ascertain the interaction mechanism at the molecular level, we further conducted molecule docking simulations based on the well-known structure of HI monomer[59] as well as the SXRD-resolved structure of *D(L)*-Cu-SMOHs. For simplification, we consider the binding conformations associated with the three lowest energies for *D*-Cu-SMOH and *L*-Cu-SMOH. The corresponding binding energies are sequentially calculated to be −10.26, −7.58, and −6.74 kJ mol⁻¹ for *D*-Cu-SMOH, which are coherently lower than the −8.75, −4.86 and −4.69 kJ mol⁻¹ for *L*-Cu-SMOH (Fig. 6a and Supplementary Table 20). In good accordance with experimental results, our theoretical simulations also claim the stronger binding between *D*-Cu-SMOH with HI. The hydrophobic amino acid residues of HI, mainly including Y16B site of α-helix and FFY segment, are

purposely orange-highlighted to present relevant chirality-dependent binding sites between HI and *D(L)*-Cu-SMOHs (Fig. 5c). Taking the lowest energy binding conformations for representation, we can easily find that *D(L)*-Cu-SMOHs approach HI principally via the aforementioned hydrophobic regions with an obviously larger contact area in the case of *D*-Cu-SMOH compared with *L*-Cu-SMOH (Fig. 6c, f). In detail, *D*-Cu-SMOH aligns its helical chain in an approximately parallel way to the α-helix region of HI via π-π stacking at the Y16B site (4.1 Å in center-to-center distance, Fig. 6b). Although *L*-Cu-SMOH forms a similar π-π stacking (5.3 Å in center-to-center distance, Fig. 6e), it orients approximately perpendicular to the α-helix region, thereby resulting in a much smaller interaction area compared to the *D*-enantiomer. The projected interaction schematics shown in Supplementary Fig. 54 also illustrate that *D*-Cu-SMOH can more fully extend its helical conformation for a better interaction match with HI. Besides, the typical FFY region of HI also takes part in hydrophobic π-π stacking with pyridine rings from *D*-Cu-SMOH and *L*-Cu-SMOH (Fig. 5d, g). Significantly, *D*-Cu-SMOH holds a center-to-center distance of 6.0 Å, which is also closer to the center-to-center distance of 6.7 Å occurring in *L*-Cu-SMOH. In short, the molecule docking study depicts the significantly enantioselective interactions between HI and *D(L)*-Cu-SMOHs at the molecular level and corroborates conclusions drawn from ITC and FL quenching experiments.

## Discussion

In this study, we developed an assembly-disassembly strategy to construct two types of copper-based single-unit metal-organic helices featuring mirror-correlated chiral structures. The resulting $D(L)$-Cu-SMOHs have been comprehensively characterized in terms of chiroptical properties, coordination structures, electronic configurations and helicities. By taking advantage of their fully exposed single-unit structures, $D(L)$-Cu-SMOHs both have exhibited high inhibition efficiencies on amyloid fibrillization, in which an enantioselectively higher performance was acquired by the $D$-Cu-SMOH. Systematic thermodynamic studies claim an entropy-favored interaction mechanism between $D(L)$-Cu-SMOHs and amyloid protein, primarily driven by hydrophobic $\pi$-$\pi$ stacking. According to the molecule docking simulation, $D$-Cu-SMOH is considered to more tightly approach the hydrophobic $\alpha$-helix region and FFY segment, which are of essential amyloidogenesis, hence unraveling the enantioselective inhibition mechanism. In light of our results, the proposed assembly-disassembly protocol is available for constructing chiral framework nanomaterials, which could have applications beyond biomaterials, including asymmetric catalysis, chiral detection, chiroptics, and chirality-dependent spin devices.

## Methods

### Materials

Cupric acetate anhydrous [$Cu(CH_3COO)_2$, 98%], sodium hydroxide (NaOH, 99%), $L$-threonine (99%), $D$-threonine (98%), glycine (99%), $D_2O$ (99.9%), methanol (MeOH, analytical grade), and ethanol (EtOH, analytical grade) were purchased from Beijing Innochem Science & Technology Co., Ltd. Sodium borohydride ($NaBH_4$, 97%) was obtained from Tianjin Kermel Chemical Reagent Co., Ltd. Concentrated hydrochloric acid (HCl, analytical grade) and acetone (99.5%) were supplied by Tianjin Fengchuan Chemical Reagent Co., Ltd. Human insulin was obtained from Shanghai Yuanye Bio-Technology Co., Ltd. ThT and 4-pydidinecarboxaldehyde (98%) were commercially available from Shanghai Aladdin Biochemical Technology Co., Ltd. Sodium chloride (NaCl) was purchased from Sinopharm Chemical Reagent Co., Ltd. Thiazolyl blue tetrazolium bromide (MTT) was purchased from Sigma-Aldrich Trading Co., Ltd. Deionized (DI) water (18 MΩ cm) used in our experiments was obtained from the laboratory water purification system. All chemicals were used directly without further purification.

### Synthesis of $D(L)$-Py-Thr

Typically, $D(L)$-threonine (2.02 g, 17.0 mmol) and NaOH (0.68 g, 17.0 mmol) were dissolved in 10 mL of DI water. Subsequently, a 10 mL methanolic solution containing 4-pyridine formaldehyde (1.82 g, 17.0 mmol) was added dropwise over 10 min at 25 °C with vigorous stirring. Continuing the reaction at 25 °C for 12 h, the solution turned yellow and was transferred to an ice water bath. Immediately, $NaBH_4$ (0.8 g, 20 mmol) dissolved in 10 mL of ice water was added in two portions at an interval of 2 h. The resultant solution was stirred for an additional 7 h and was then adjusted to a pH value of 5.5 using 1.0 M HCl. After stirring at 25 °C for an additional 12 h, the crude product was collected by rotary evaporation of the solvent. The final product was extracted with hot ethanol (3 × 150 mL). The combined ethanolic extracts were evaporated, and the resultant solid was washed with 10 mL of acetone. Finally, a white product was obtained for further characterization and use.

Yield: 2.39 g (67.6%) for the $D$-enantiomer. $^1$H-NMR ($D_2O$, ppm): -CH$_3$ (1.19, d, 3H), -HN-CH (3.28, d, 1H), -CH (3.96, m, 1H), -CH$_2$ (4.21, dd, 2H), py-H (7.46, d, 2H), py-H (8.52, d, 2H). $^{13}$C-NMR ($D_2O$, ppm): -CH$_3$ (19.68), -CH$_2$ (49.16), -CH-OH (66.28), -CH-HN (68.45), py-C (125.18), py-C (141.52), py-C (149.12), -COOH (171.54). FT-IR (KBr, cm$^{-1}$): $v_{OH}$, 3480; $v_{NH}$, 2965; $v_{as}(CO_2)$, 1602; $v_s(CO_2)$, 1421. The observed mass spectrum of protonated d-threonine showed $m/z$: 211.11 (100%); $m/z$: 212.11 (11.3%). Calculated values: $m/z$: 211.11 (100%); $m/z$: 212.11 (11.8%).

Specific rotation value: +24.5°. The NMR, MS, FT-IR, and specific rotation characterization results are provided in Supplementary Figs. 1, 3, 5, 7, and Supplementary Table 1, respectively.

Yield: 2.45 g (69.1%) for the $L$-enantiomer. $^1$H-NMR ($D_2O$, ppm): -CH$_3$ (1.21, d, 3H), -HN-CH (3.28, d, 1H), -CH (3.96, m, 1H), -CH$_2$ (4.22, dd, 2H), py-H (7.48, d, 2H), py-H (8.52, d, 2H). $^{13}$C-NMR ($D_2O$, ppm): -CH$_3$ (19.64), -CH$_2$ (49.11), -CH-OH (66.24), -CH-HN (68.45), py-C (125.16), py-C (141.45), py-C (149.09), -COOH (171.46). FT-IR (KBr, cm$^{-1}$): $v_{OH}$, 3482; $v_{NH}$, 2967; $v_{as}(CO_2)$, 1602; $v_s(CO_2)$, 1421. Obtained $m/z$: 211.11 (100%); $m/z$: 212.11 (11.8%). Calculated $m/z$: 211.11 (100%); $m/z$: 212.11 (11.8%). Specific rotation value: −25.3°. The NMR, MS, FT-IR, and specific rotation characterization results are provided in Supplementary Figs. 2, 4, 6, 7, and Supplementary Table 1, respectively.

### Synthesis of Py-Gly

Similarly, glycine (1.28 g, 17.0 mmol) and NaOH (0.68 g, 17.0 mmol) were dissolved in 10 mL of DI water. Subsequently, a 10 mL methanolic solution containing 4-pyridine formaldehyde (1.82 g, 17.0 mmol) was added dropwise over 10 min at 25 °C with vigorous stirring. Continuing the reaction at 25 °C for 12 h, the solution turned yellow and was transferred to an ice water bath. Immediately, $NaBH_4$ (0.8 g, 20 mmol) dissolved in 10 mL of ice water was added in two portions at an interval of 2 h. The resultant solution was stirred for an additional 7 h and was then adjusted to a pH value of 6.5 using 1.0 M HCl. After stirring at 25 °C for an additional 12 h, the crude product was collected by rotary evaporation of the solvent. The final product was extracted with hot ethanol (3 × 150 mL). The combined ethanolic extracts were evaporated, and the resultant solid was washed with 10 mL of acetone. Finally, a white product was obtained for further characterization and use.

Yield: 2.14 g (75.4%) for the Py-Gly. $^1$H-NMR ($D_2O$, ppm): −HN−CH$_2$ (3.64, s, 2H), −CH$_2$ (4.29, s, 2H), py-H (7.51, d, 2H), py-H (8.56, d, 2H). $^{13}$C-NMR ($D_2O$, ppm): −CH$_2$ (49.49), −CH$_2$-HN (49.25), py-C (124.98), py-C (141.27), py-C (149.65), −COOH (171.36). FT-IR (KBr, cm$^{-1}$): $v_{OH}$, 3428; $v_{NH}$, 3039; $v_{as}(CO_2)$, 1608. The observed mass spectrum of protonated glycine showed $m/z$: 167.08 (100%); $m/z$: 168.08 (9.0%). Calculated $m/z$: 167.08 (100%); $m/z$: 168.09 (8.9%). The NMR, MS, FT-IR, and specific rotation characterization results are provided in Supplementary Figs. 45–48, respectively.

### Assembly of $D(L)$-Cu-crystals single-crystals

In a typical synthesis, $D(L)$-Py-Thr ligand (31.50 mg, 0.15 mmol) was dissolved in a mixed solvent containing 1.7 mL DI $H_2O$ and 0.8 mL methanol. To prevent precipitating upon metal precursor addition, the solvent pH value was pre-adjusted to 4.0 using 1.0 M aqueous HCl. $Cu(CH_3COO)_2$ (54.75 mg, 0.30 mmol) was added subsequently under vigorous stirring, followed by ultrasonication for 2 min to ensure complete dissolution. The pH value of the reaction system was further adjusted to 3.2 using 1.0 M aqueous HCl to slow down the growth rate during single-crystal cultivation. Finally, the cultivation process was initiated at a 60 °C oven under static conditions for a week to produce blue rectangular $D(L)$-Cu-crystals suitable for X-ray diffraction analysis.

### Assembly of $D(L)$-Cu-crystals powder

Scaled-up production was accomplished through simply increasing the concentration of precursors. Specifically, $D(L)$-Py-Thr (42 mg, 0.2 mmol) was dissolved in a mixed solvent of 1.7 mL DI $H_2O$ and 0.8 mL methanol. To prevent precipitation, the pH value of the above solution was pre-adjusted to 4.0 using 1.0 M HCl prior to the addition of copper(II) acetate. Subsequently, $Cu(CH_3COO)_2$ (73 mg, 0.4 mmol) was added to the solution and the mixture was sonicated for 2 min to full dissolution. The resultant transparent solution was further acidified to pH of 3.23 using 1.0 M HCl solution and was transferred into a sealed 5 mL glass vial for three days at 60 °C. Finally, blue powders were obtained by a simple filtration method.

Yield: 41.9 mg, 63.5% for the *D*-enantiomer. Sizes: an average length and width of 15.9 μm and an average height of 4.3 μm. Yield: 42.3 mg, 64.1% for the *L*-enantiomer. Sizes: an average length and width of 16.1 μm and an average height of 4.1 μm. The SEM characterization results are available in Supplementary Figs. 10, 11, respectively.

## Disassembly construction of *D(L)*-Cu-SMOHs

Prior to disassembly, *D(L)*-Cu-crystals were pretreated with mechanical milling for 5 min to reduce particle sizes. Subsequently, 5 mg of milled powders were dispersed in 5 mL of DI $H_2O$ under vigorous stirring (800 rpm). The resultant suspension was followed by ultrasonication treatment for 2 min and then was placed in a shaker (120 rpm) maintained at a constant temperature of 60 °C for 60 min. Finally, the colloidal dispersion of disassembled *D(L)*-Cu-SMOHs was obtained.

## Reassembly of *D(L)*-Cu-SMOHs

Reassembly of the *D(L)*-Cu-SMOHs framework was executed by simply introducing an aprotic solvent. Specifically, ACN was added dropwise as the aprotic solvent to the aqueous dispersion of *D(L)*-Cu-SMOHs until achieving a 1:1 v/v ratio of ACN: $H_2O$. After 30 min of static incubation, blue precipitates formed and were isolated by centrifugation (2570 × g, 5 min) for subsequent characterization.

## Characterization

SXRD data were collected at 100.1(3) K using a XtaLAB AFC12 (RINC): Kappa single diffractometer (Cu $K_\alpha$ radiation, $\lambda = 1.542$ Å). The SXRD data reduction and structure solution were carried out using the OLEX2 (ver. 1.5) software suite. PXRD patterns were acquired on a Rigaku Ultima IV X-ray diffractometer (Cu $K_\alpha$, $\lambda = 1.542$ Å) equipped with a panel detector, operating at 40 kV and 30 mA. Single-crystal morphology was obtained using a Tianjin Laike Optics LK200M microscope. FT-IR spectra were recorded on an FTIR-650 (Gangdong Sci.&Tech. Co., Ltd, Tianjin, China) using the KBr tablet method. TGA was performed on a Netzsch TG 209 F3 Tarsus amid nitrogen flow ($N_2$, 20 mL min$^{-1}$) with a heating rate of 10 °C min$^{-1}$ from 40 °C to 840 °C. The FL was conducted using a Gangdong F-320 fluorescence spectrophotometer. CD spectra were acquired on an Applied Photophysic Chirascan V100 (Plus) spectrometer. Nuclear magnetic resonance (NMR) was performed on a Bruker AVANCE III HD 400 machine. Mass spectrometry (MS) was obtained on a Shimadzu LCMS-8040 machine. HR-TEM was carried out on a JEM-F200 at a voltage of 200 KV. Scanning electron microscope (SEM) measurements were carried out on a Phenom ProX machine at 5.0 kV. The specific rotation was measured on a Shanghai Shenguang SGWzz$^{-2}$ instrument at 589 nm (i.e., the *D* line of sodium). DLS measurement was performed on a Malvern Zetasizer Pro. Cell viability assessment via MTT was measured by a microplate absorbance reader (VICTOR Nivo™, PerkinElmer). AFM images were recorded using a Multimode-8 AFM (Bruker, USA) in Smartmode with 512 × 512 pixels resolution and 1.0–1.5 Hz scan rate. Image processing was conducted using NanoScope Analysis 1.40 software. TEM analysis was conducted using a Hitachi H7650 at 100 kV. Fluorescence images were acquired with an Olympus FV3000 confocal laser scanning microscope. VCD spectra were acquired on a ChiralIR-2X spectrometer.

## EPR measurements and coordination structure identifications

The Bruker Magnettech ESR5000 EPR spectrometer was used to measure the EPR spectra of samples at room temperature (25 °C) with a magnetic field strength scanning from 200 to 400 mT and a speed of 10 mT s$^{-1}$. The modulation frequency of the implemented microwave is 100 kHz. The simulation of the obtained spectral lines was performed using Bruker Xenon software. The spin Hamiltonian formalism was employed to interpret the anisotropic $g_i$ deviations. According to the perturbation theory, the spin-orbit coupling (SOC) between the ground

state $|0\rangle$ and excited state $|n\rangle$ reintroduces a small orbital contribution to the $g_i$ value deviating from a typical $g_e$ of 2.0023[36]. That is: $g_i = g_e + 2\lambda \sum_{n=1,2,\ldots} \left( \frac{\langle 0|\hat{L}_i|n\rangle\langle n|\hat{L}_i|0\rangle}{E_0-E_n} \right) = 2.0023 + \lambda \sum_{n=1,2,\ldots} \left( \frac{2\langle 0|\hat{L}_i|n\rangle\langle n|\hat{L}_i|0\rangle}{E_0-E_n} \right)$. Where $i = x$, $y$, or $z$, indicating the anisotropic SOC effect depending on the coordination geometry of central metal ions, $\lambda$ is the SOC constant, $\hat{L}_i$ is the operator of orbital angular momentum, $E_0$, and $E_n$ are the energies of the ground and the excited states, respectively. In our experiments, $|n\rangle$ and $E_n$ are the 3d orbitals and corresponding energy level of Cu(II) featuring an unpaired $3d^9$ configuration, respectively. Supplementary Table 15 shows all the integrals for 3d orbitals and only 16 of them result in non-zero values. So the item of $2\langle 0|\hat{L}_i|n\rangle\langle n|\hat{L}_i|0\rangle$ can only take values from the numerical group of 2, 6, and 8, which can be further identified according to the magic pentagon rule (Supplementary Fig. 22)[33].

According to the octahedron field theory, the 3d ground state of Cu(II) with occupation of one unpaired electron is only the case of the $3d_{x^2-y^2}$ or $3d_{z^2}$ orbital. Hence, the EPR characteristic of Cu(II) is axial symmetry with two Landé *g*-factors[36]: $g_\parallel(g_z)$ and $g_\perp(g_x = g_y)$. When the coordination geometry is trigonal bipyramidal, the ground state is the $d_{z^2}$ orbital based on the Jahn–Teller effect. Hence, $g_\perp = g_x = g_y = 2.0023 - \frac{6\lambda}{E(d_{z^2})-E(d_{xz})} = 2.0023 - \frac{6\lambda}{E(d_{z^2})-E(d_{yz})} = 2.0023 - \frac{6\lambda}{\Delta E_{1,2}}$ and $g_\parallel = g_e = 2.0023$, where $\Delta E_{1,2}$ is the energy gap between the ground-state $d_{z^2}$ and the $d_{xz}$ or $d_{yz}$ orbitals. The $\lambda$ vaule for Cu(II) is −830 cm$^{-1}$, therefore a corresponding EPR of trigonal bipyramidal coordination is characterized by $g_\perp > g_\parallel \geq g_e = 2.0023$[69]. When the geometry is elongated octahedral, the ground state is the $d_{x^2-y^2}$ orbital[35,70], $g_\perp = g_x = g_y = 2.0023 - \frac{2\lambda}{E(d_{x^2-y^2})-E(d_{yz})} = 2.0023 - \frac{2\lambda}{E(d_{x^2-y^2})-E(d_{xz})} = 2.0023 - \frac{2\lambda}{\Delta E_{1,2}}$ and $g_\parallel = g_z = 2.0023 - \frac{8\lambda}{E(d_{x^2-y^2})-E(d_{xy})} = 2.0023 - \frac{8\lambda}{\Delta_3}$. Where $\Delta E_{1,2}$ is the energy gap between the ground-state $d_{x^2-y^2}$ and the $d_{xz}$ or $d_{yz}$ orbitals, while $\Delta E_3$ is the energy gap between the ground-state $d_{x^2-y^2}$ and the $d_{xy}$ orbital. In these cases, the relation $g_\parallel > g_\perp > g_e = 2.0023$ is expected.

## Structure modeling and geometry optimization of *D*-Cu-SMOH

The structure of *D*-Cu-SMOH was built by isolating a one-dimensional (1D) helical chain from the unit cell of *D*-Cu-crystal using the Material Studio (ver. 19.1) software suite. To account for potential coordination of water molecules during the disassembly process, an additional $H_2O$ molecule was introduced to the Cu(II) center in *D*-Cu-SMOH. Then, geometry optimization was performed using the Forcite module integrated in the Material Studio package at the ultrafine quality level.

## XAFS measurements

XAFS measurements at Cu *K*-edge (8979 eV) were carried out at the BL11B beamline of the Shanghai Synchrotron Radiation Facility (SSRF), whose electron storage ring was operated at 3.5 GeV with a beam current of 180 mA. A Si (111) double-crystal monochromator was used, with the energy calibrated using the standard Cu foil. All measurements were performed at room temperature (25 °C) using transmission mode with two ionization chambers filled with $N_2$ gas to record the intensity of incident and transmitted X-ray. The collected data were processed and analyzed using the Demeter (ver. 0.9.26) software package for Windows[71]. Specifically, the data were processed using the Athena (ver. 0.9.26) software, by calibration with the standard Cu foil as the reference and subtraction of a linear pre-edge and normalization by the edge jump. The $\chi(k)$ data were isolated by subtracting a smooth, third-order polynomial approximating the absorption background of an isolated atom. The data were $k^3$-weighted prior to the Fourier transformation. To obtain the quantitative structural parameters

around the central atoms (coordination number, distance to the scattering atom, Debye-Waller factor and $E_0$ shift), a least-squares curve-fitting analysis of the Fourier-transformed data in $R$-space was carried out using the Artemis (ver. 0.9.26) software.

## Pretreatments of HI

The stock solution of human insulin (HI) was prepared by dissolving the protein in 1,1,1,3,3,3-hexafluoroisopropanol (HFIP) at a concentration of 3 mg mL$^{-1}$. To ensure complete disaggregation of pre-existing oligomers, the solution was incubated overnight at ambient temperature (25 °C) under constant agitation (120 rpm). Residual HFIP was subsequently removed via N$_2$ stream purging. The purified monomeric HI was stored in a −20 °C refrigerator before further experimentation.

## HI fibrillization experiments

The fibrillization experiment was prepared via diluting the HI stock solution in aqueous solution (pH = 2.0) containing 0.1 M NaCl to accelerate HI fibrillization[43]. The obtained HI (200 μg mL$^{-1}$, 34 μM) incubation solution was initiated at 60 °C for 6 h, either in the absence or presence of $D(L)$-Cu-SMOHs at concentrations of 0, 12.5, 25, 50, 100, and 200 μg mL$^{-1}$, respectively. Aliquots were collected at predetermined intervals and flash-frozen at −20 °C for subsequent analysis. ThT fluorescence, CD, and FT-IR spectroscopy were employed to monitor HI fibrillization kinetics.

## ThT fluorescence spectroscopy

ThT probe was employed to monitor the amyloid fibrillization kinetics of HI monomers incubated at 60 °C accompanied by an increase in FL intensity[59]. After incubation for a certain duration, 100 μL of HI solution incubated with or without $D(L)$-Cu-SMOHs was periodically withdrawn and mixed with 2 mL of 50 μM ThT using vortex rotation. After an equilibrium of 2 min, the FL spectrum was recorded using a Gangdong F-320 fluorescence spectrophotometer. The excitation wavelength was set at 450 nm, and the FL intensity was monitored at an emission peak of 485 nm. Each data point was performed in triplicate and the averaged FL intensity was further fitted as a function of incubation time via the sigmoidal model[44].

## Stopped-flow experiments

Stopped-flow fluorescence experiments were performed using an SX20 stopped-flow spectrometer (Applied Photophysics Ltd., UK). Kinetic data analysis was conducted through the integrated Pro-Data (ver. 4.5) software package provided with the instrument. Fluorescence emission was monitored at 302 nm under excitation of 278 nm.

## CD study of secondary structures of HI

CD spectra (Chirascan V100 Plus, Applied Photophysics) were employed to study secondary structure transitions during the fibrilization process of HI in the absence or presence of $D(L)$-Cu-SMOHs. Experimental data were collected over the range of 195 to 260 nm at a scanning rate of 100 nm min$^{-1}$. The CD spectra of all samples were collected by averaging data from three successive scans. A Neural Network method using CDNN (ver. 2.1) software was adopted to calculate the proportion of secondary structures.

## Cell cytotoxicity assay

The cytotoxicity of $D(L)$-Cu-SMOHs was quantitatively evaluated using the classical MTT colorimetric assay. Specifically, SH-SY5Y human neuroblastoma cells were cultured in Dulbecco's minimum essential medium (DMEM) in a 96-well round-bottom plate with a density of 1 × 10$^4$ cells per well under a temperature of 37 °C and an atmosphere of 5% CO$_2$. After incubation for 24 h, the culture medium was replaced with 200 μL of fresh DMEM containing $D(L)$-Cu-SMOHs with varying concentrations (0, 3, 6, 13, 25, 50, and 100 μg mL$^{-1}$). Continuing for an additional 24 h coincubation period, apoptotic cells were rinsed with

pre-warmed PBS solution (pH = 7.4). Subsequently, 100 μL of MTT reagent (0.5 mg mL$^{-1}$) dissolved in DMEM was introduced to each well. After staining for 4 h, 100 μL of DMSO was added to dissolve the precipitated MTT formazan. The samples' absorbance at 490 nm was measured for each well using an automatic microplate reader (VICTOR Nivo™). Three independent experiments were carried out for each sample and the averaged data were taken for analysis and discussion.

## Thermodynamics by ITC

Nanowatt ITC was performed using a Nano ITC instrument (TA Instruments, Inc.) to quantitatively measure the thermodynamic parameters associated with interactions between $D(L)$-Cu-SMOHs and HI at 298.15 K. In a typical ITC experiment, 250 μL of $D(L)$-Cu-SMOHs solution (3.03 mM) was injected into the sample cell containing 1 mL of HI solution (0.17 mM) via a step of 10 μL per injection. Similarly, 250 μL of $D(L)$-Py-Thr (3.03 mM), Py-Gly (3.03 mM), and Cu-Gly-MOF (3.03 mM) solutions, respectively, were injected into the sample cell containing 1 mL of HI solution (0.17 mM) via a step of 10 μL per injection. The mixed solution in the sample cell was continuously stirred at 250 rpm throughout the titration process. A 600-s interval was set between successive injections to ensure a complete thermal equilibration for each titration. The titration heat namely enthalpy change ($\Delta H$) was directly measured after titration. The association constant ($K_a$) and binding stoichiometry ($n$) were obtained by fitting collected thermograms using an independent binding model on NanoAnalyze software (TA Instruments Inc.). The correlated Gibbs free energy change ($\Delta G$) was calculated based on the relationship of $\Delta G = -RT\ln K_a$ (R is the ideal gas constant, and $T$ = 298.15 K). Finally, the associated entropy change ($\Delta S$) was determined by the equation of the $\Delta G = \Delta H - T\Delta S$.

## Temperature-dependent FL experiments

Fluorescence quenching experiments were performed using a Gangdong F-320 fluorescence spectrophotometer equipped with a digital circulating water bath for temperature control at 298.15, 304.15, 310.15 and 316.15 K. The excitation wavelength was set at 278 nm, and the emission spectrum was recorded from 290 to 400 nm. The FL quenching data by addition of $D(L)$-Cu-SMOHs were fitted as a function of concentration using the modified Stern-Volmer equation[68]: $\frac{F_0}{\Delta F} = \frac{1}{f_a K_a}\frac{1}{[M]} + \frac{1}{f_a}$. Where $F_0$ represents the initial fluorescence of HI, $\Delta F$ represents the difference in fluorescence intensity after adding $D(L)$-Cu-SMOHs, $K_a$ is the apparent binding constant, $[M]$ is the concentration of added $D(L)$-Cu-SMOHs, $f_a$ is the fraction of accessible fluorophore. As $K_a$ is temperature dependent, $\Delta H$ and $\Delta S$ can be obtained using the van 't Hoff equation[56]: $\ln K_a = -\frac{\Delta H}{RT} + \frac{\Delta S}{R}$. In which, $T$ is the tested temperature, R is the ideal gas constant. Corresponding $\Delta G$ at certain temperatures can be calculated using: $\Delta G = -RT\ln K_a$.

## Molecule docking studies

Molecular docking was conducted to identify potential binding interactions between $D(L)$-Cu-SMOHs on HI monomer using the AutoDock (ver. 4.2) software[48]. The 3D structure model of $D(L)$-Cu-SMOHs was built according to the SXRD data. While the native structure of HI can be drawn from the RCSB Protein Data Bank (PDB ID: 1GUJ), which is a dimeric structure consisting of two identical HI monomers, but in an antiparallel alignment fashion. (i.e., the parallel HI monomer has the A and B chains, and the antiparallel one has the C and D chains)[59]. Therefore, we only take the HI monomer consisting of A and B chains as the structure model. A blind docking strategy was applied using a grid box dimensioned to 98, 100, and 94 points along the $X$, $Y$, and $Z$ axes, respectively, with a grid spacing of 0.375 Å. Prior to docking, Gasteiger partial charges were assigned to both the HI monomer and $D(L)$-Cu-SMOHs. The autodocking conformations with three minimal binding affinities were analyzed and visualized using PyMOL (ver. 2.3.0) (The PyMOL

**Article** https://doi.org/10.1038/s41467-025-63459-2

Molecular Graphics System, Version 2.0 Schrödinger, LLC.)[72] for 3D representations and LigPlot+ (ver. 2.2) software for 2D interaction diagrams.

## Reporting summary

Further information on research design is available in the Nature Portfolio Reporting Summary linked to this article.

## Data availability

The data that support the findings of this study are available from the corresponding authors upon reasonable request. The crystallographic data generated in this study has been deposited in the Cambridge Crystallographic Data Centre (CCDC) with deposition numbers of 2330456 and 2330457. CIFs are provided as Supplementary Data 1, 2. Source data are provided with this paper.

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

## Acknowledgements

The authors acknowledge financial support from the National Natural Science Foundation of China (Grant No. 22103055, J.G. and 22473083, M.T.Z.), Natural Science Foundation of Tianjin (Grant No. 24JCZDJC00190, J.G.), Hebei Natural Science Foundation (Grant No. B2025110011, J.G.), Excellent Scientific Special Commissioner Foundation of Tianjin (Grant No. 24YDTPJC00640, J.G.) and PEIYANG Young Scholars Program of Tianjin University (Grant No. 2020XRX-0023, M.T.Z.) and the Project Supported by Hebei Technological Innovation Center of Chiral Medicine (Grant No. ZXJJ20240201, M.T.Z. and J.G.). We thank the Analytical & Testing Center of Tiangong University for the help of HR-TEM testing. We thank the staff at SSRF BL11B beamline for providing technical support in X-ray absorption fine structure data collection.

## Author contributions

J.G. proposed the research concept, Y.L.J., and C.Y.Y. performed the main experiments, Y.T.Y., T.T.Z., S.Y.K., Z.L.Z., Y.L.L., J.L., R.X.M., and Z.Y.B. analyzed the data, H.L.C., P.L., K.Y., and Z.Y.T. participated in the discussion and supervised the experiments, Y.Z, M.T.Z., Y.L and J.G. co-wrote the manuscript.

## Competing interests

The authors declare no competing interests.
