## [Transparent Peer Review file · Nature Communications]

Disassembly of chiral hydrogen-bonded frameworks into single-unit organometallic helices for enantioselective amyloid inhibition

Corresponding Author: Professor Jun Guo

Version 0:

Reviewer comments:

Reviewer #1

(Remarks to the Author)

The authors describe here chiral Cu(II) complexes based on N-(4-pyridylmethyl)-d(l)-threonine ligands which primarily arrange in helical coordination chains that interact through hydrogen-bonding with neighboring chains to provide a hydrogen-bonded network in the crystal. A whole battery of techniques, besides SCXRD, was used to characterize the crystalline compounds and also the fibrils resulting from the disassembly of the hydrogen-bonding network. The authors then investigated the inhibition activity of the disassembled fibrils on human insulin (HI) fibrillization, HI being a representative of amyloid proteins. The D(L)-Cu-SMOHs fibrils showed an inhibition activity on the HI fibrillization of up to 40% as determined by thioflavin T fluorescence probe, this process being also enantioselective. Indeed, stronger binding has been found both by experiment and by molecular docking simulations between D-Cu-SMOHs and HI than between L-Cu-SMOHs and HI. With this being said, while I acknowledge the scientific aspects of the work and the robustness of the investigations, I consider the study lacks of sufficient novelty for publications in Nat. Commun. The main reason of my criticism comes from the fact that the authors published last year (2024) in Nat. Commun. (ref. 14) exactly the same approach for the synthesis of such crystalline chiral hydrogen-bonded architecture with the same N-(4-pyridylmethyl)-d(l)-threonine ligands. The synthesis of the latter was already thoroughly described in ref. 14, together with Zn(II) coordination polymers showing the same property of disassembling in fibrils. In the present report the authors describe again, at the same level of details, the ligands already reported, without clearly mentioning the close vicinity of the investigations in ref. 14 and in the present report.

Reviewer #2

(Remarks to the Author)

In this work from Guo and co-workers, the authors report enantiopure helical copper -based coordination polymers and their effect on amyloid fibrillization. To keep it short, I find the reported work very interesting for a broad audience and well characterized, certainly worth publishing in Nature Communications. The results are supported by several well performed complementary analyses, each well described in a scholar manner, with an abundance of explanations and details.

I have only a few small concerns and comments regarding this work.

- 1) Is the assembly-disassembly of the chiral coordination polymers reversible? From how it is depicted in Scheme 1 it should be reversible, however from the results there is no evidence of this. It seems that upon breaking the hydrogen-bonded framework it's not possible to go back. The authors should verify this, for instance supported by PXRD, otherwise scheme 1 should be changed accordingly.
- 2) The PXRD spectra are calculated from SCXRD and not simulated. No computational method is employed but are directly calculated from the structure factors obtained from SCXRD, please correct this inaccuracy.
- 3) The CD spectra of the D and L-Cu-SMOH are well reported, however why no absorption spectra are reported? These should be added, also to appreciate more the strong gabs arising in the metal d-d transitions area.
- 4) In the main text at page 24 row 312 the authors write: "Apparently, the retained CD intensity by D-Cu-SMOH is also stronger than that of L-Cu-SMOH, further demonstrating the higher enantioselective inhibition effect of D-Cu-SMOH on HI fibrillization." However, the CD spectra in Fig. S22 are identical, while the data reported in the main, as fig. 3d and consequent TEM and AFM shows the different enantioselectivity. Is Fig. S22 reporting the correct data?
- 5) In part describing the interaction with amyloid, no control experiments with the ligands (L)-Py-Thr are reported. Why?

Reviewer #3

(Remarks to the Author)

The manuscript by Tang and co-workers introduces a biomimetic “assembly–disassembly” route to fabricate enantiopure, single unit Cu–organic helices whose chirality is preserved upon water triggered disassembly. The authors show that D and L helices inhibit insulin amyloid formation in vitro. The concept is interesting, but some control experiments will be useful to solidify the central findings of the work.

- Entropy gains can also arise from release of structured water, counter ion effects or ligand reorganization. Thereby, the claim that $\Delta H > 0$ and $\Delta S > 0$ unambiguously prove a hydrophobic π – π stacking mechanism can be strengthened by more experiments. Entropy gains can also arise from release of structured water, counter ion effects or ligand reorganization. Some controls experiments (i.e. ITC of free D(L)-Py-Thr, achiral Cu-framework built with a non-aromatic ligand, a pyridyl ligand lacking chirality) will help to understand these contributions.
- The paper will benefit of a demonstration, through control experiments, that the chiral pyridyl threonine ligand or an achiral Cu framework alone do not exert similar anti amyloid effects.
- Structural characterization (CD, DLS, TEM) should be also performed under physiological conditions.
- The work will benefit of the quantification through vibrational circular dichroism or Raman optical activity analysis of the helicity and chiral density of SMOHs, and their correlation to inhibitory potency.

Version 1:

Reviewer comments:

Reviewer #1

(Remarks to the Author)

I appreciate the authors' responses and their efforts to improve the quality of the manuscript and to provide additional experimental proofs to support their hypotheses and conclusions. I am still not totally convinced that this work should be published in Nat. Commun. since a very similar work by the same authors (ref 14) has been published in Nat. Commun. in 2024, but let's say this revised manuscript could be published.

Reviewer #2

(Remarks to the Author)

In this revised submission of the work from Guo and co-workers, the authors answered all my questions in a satisfactory manner. I believe that the extra experiment added improved the manuscript further, and my previous revision was already positive. The assembly/disassembly process is proven in a smart way, using acetonitrile as a solvent and well supported by PXRD and SEM data. The updated discussion on the absorption data matches well the EPR data. My concern about CD experiments has been clarified in both the main text and supporting information. Finally, the control experiments with only the ligand add a solid extra experiment to the overall findings.

Despite the similarity with the author's previous work with Zn(II), in this manuscript the findings and characterization is interesting enough and fits the scope of the journal.

Therefore, I believe this revised version is worth publishing as it is in Nature Communications.

Reviewer #3

(Remarks to the Author)

The authors have addressed all the concerns raised in the previous round and the paper is now suitable for publication.

Response to Reviewers' Comments

Reviewer #1

General comments:

The authors describe here chiral Cu(II) complexes based on N-(4-pyridylmethyl)-d(l)-threonine ligands which primarily arrange in helical coordination chains that interact through hydrogen-bonding with neighboring chains to provide a hydrogen-bonded network in the crystal. A whole battery of techniques, besides SCXRD, was used to characterize the crystalline compounds and also the fibrils resulting from the disassembly of the hydrogen-bonding network. The authors then investigated the inhibition activity of the disassembled fibrils on human insulin (HI) fibrillization, HI being a representative of amyloid proteins. The *D(L)*-Cu-SMOHs fibrils showed an inhibition activity on the HI fibrillization of up to 40% as determined by thioflavin T fluorescence probe, this process being also enantioselective. Indeed, stronger binding has been found both by experiment and by molecular docking simulations between *D*-Cu-SMOHs and HI than between *L*-Cu-SMOHs and HI. With this being said, while I acknowledge the scientific aspects of the work and the robustness of the investigations, I consider the study lacks of sufficient novelty for publications in Nat. Commun. The main reason of my criticism comes from the fact that the authors published last year (2024) in Nat. Commun. (ref. 14) exactly the same approach for the synthesis of such crystalline chiral hydrogen-bonded architecture with the same N-(4-pyridylmethyl)-d(l)-threonine ligands. The synthesis of the latter was already thoroughly described in ref. 14, together with Zn(II) coordination polymers showing the same property of disassembling in fibrils. In the present report the authors describe again, at the same level of details, the ligands already reported, without clearly mentioning the close vicinity of the investigations in ref. 14 and in the present report.

Response: We thank this reviewer for raising critical points regarding the novelty and advancement of our current manuscript relative to prior work. As also noted by the reviewer, the primary contribution of Ref. 14 was the construction and structural characterization (via SXR) of novel chiral hydrogen-bonded frameworks employing

the *N*-(4-pyridylmethyl)-*d*(*l*)-threonine ligand with a Zn(II) precursor. However, fundamental gaps and unresolved challenges persisted in Ref. 14: I) While suggesting potential disassembly behavior, Ref. 14 didn't systematically investigate or characterize the disassembly products of chiral hydrogen-bonded framework crystals due to a lack of validated methodologies for such analysis. II) The anticipated inheritance of helicity from the chiral frameworks to the disassembled fibrils was not experimentally characterized or confirmed in Ref. 14. III) Owing to limited understanding of the disassembled structures, Ref. 14 did not report any functional properties, explore potential applications, or examine structure-performance relationships.

It is necessary to note that the novelty and advancement of this work extend beyond reporting single-unit metal-organic helices (SMOHs) disassembled from novel copper-based ***D(L)***-Cu-Crystals. More critically, this work develops a combinational methodology to unambiguously resolve atomic-level structures of disassembled SMOHs, including metal coordination geometry, electronic configurations and helicity inheritance. Leveraging dense chiral active sites and adaptive helical conformations, this work further demonstrates potent and chirality-dependent inhibition performance of disassembled ***D(L)***-Cu-SMOHs against amyloid fibrillization, and in-depth reveals the underlying structure-performance relationships through combined experimental/theoretical results. As detailed below, comprehensive discussions and supplemented experiments have been incorporated into the revised manuscript in order to rigorously substantiate our claims. It is expected that our response and revisions can satisfactorily address the concerns. We earnestly request the reviewer's reconsideration of our manuscript.

(1) This work reports novel *D(L)*-Cu-Crystals as dynamic and reversible assembly-disassembly-reassembly platforms.

Despite sharing the same ligands, it is crucial to point out that ***D(L)***-Cu-Crystals reported in this work are fundamentally different from those in Ref. 14 in terms of composition, structure, and crystallography. As compared and summarized in Table R1, ***D(L)***-Cu-Crystals show different central metal ions [Cu(II) vs. Zn(II)], different

coordination modes [trigonal bipyramidal of Cu(II) vs. elongated octahedral of Zn(II)], and distinct crystallographic cells (details in Table R1). These structural innovations are unambiguously validated by the new CCDC deposition numbers of 2330456 for *D*-Cu-Crystals and 2330457 for *L*-Cu-Crystals.

Table R1. Comparison of *D(L)*-Cu-Crystals to previous *d(l)*-Zn-HOIFs

Sample	D(L) -Cu-Crystal (This Work)	d(l) -Zn-HOIF (Ref. 14)
Empirical formula	CuC ₁₀ H ₁₅ ClN ₂ O ₄	ZnC ₁₂ H ₁₈ N ₂ O ₆
Formula weight	326.23	351.65
Metal type	Cu(II)	Zn(II)
Ligand	D(L) -Py-Thr	D(L) -Py-Thr
Coordination mode	Trigonal bipyramidal	Elongated octahedral
Crystal system	Tetragonal	Orthorhombic
1D Helicity	Quadruple screw	Double screw
Pitch/Å	35.6	18.4
Space group	P ₄ 2 ₁ 2/P ₄ 12 ₁ 2	P ₂ 12 ₁ 2 ₁
Cell parameter (a/Å)	9.12160(10)	6.14050(10)
Cell parameter (b/Å)	9.12160(10)	14.3368(2)
Cell parameter (c/Å)	35.5635(3)	18.3682(2)
Cell volume/Å ³	2959.01(7)	1617.05(4)
Z	8	4
$\rho_{\text{calc}}/\text{g}\cdot\text{cm}^{-3}$	1.465	1.444
CCDC number	2330456(2330457)	2209408(2209409)

Beyond the disassembly property, we have further demonstrated the reversible reassembly of *D(L)*-Cu-Crystals using the corresponding disassembled SMOHs. Thanks to the dynamic feature of hydrogen bonding, reassembly of *D(L)*-Cu-SMOHs is triggered by simply introducing acetonitrile (ACN), a low-dielectric-constant aprotic solvent, into aqueous *D(L)*-Cu-SMOHs solutions. On account of the desolvation effect of ACN, inter-chain hydrogen bonding reforms between *D(L)*-Cu-SMOHs and hence reassembles corresponding *D(L)*-Cu-Crystals. Upon adding 50% volume of ACN, crystalline chiral frameworks featuring rectangular morphologies have been successfully reassembled, as unequivocally confirmed by following PXRD and SEM results (Fig. R1).

In short, this work presents a pair of novel *D(L)*-Cu-Crystals as the dynamic and reversible assembly-disassembly-reassembly framework platform.

Fig. R1. Structural and morphological characterizations of *D(L)*-Cu-Reassemblies. (Note: Fig. R1a and Figs. R1b-g have also been added to the revised *Supplementary Information* as Fig. S26 and Fig. S27, respectively.)

(2) This work establishes a combinational methodology for valid characterizations of disassembled SMOHs.

Being subject to their aperiodic and ultrathin nature, inaccessible by conventional XRD and electron diffraction, the structural information of disassembled chiral fibrils was a previously unaddressable challenge. The key new advancement of this manuscript is the establishment of a combinational methodology using X-ray absorption fine structure spectroscopy (XAFS), vibrational circular dichroism (VCD), and electron paramagnetic resonance (EPR) to unambiguously define the structure of *D(L)*-Cu-SMOHs (Fig. 2 in the revised manuscript), including metal coordination environment, electronic configuration, and inherited helical architecture, etc.

Firstly, we found that water molecules triggered the coordination transformation of central Cu(II) upon disassembly according to EPR results. The five-coordinated

trigonal bipyramidal geometry of Cu(II) in bulk crystal transformed into six-coordinated elongated octahedral geometry in the disassembled SMOHs (Figs. 2a-c in the revised manuscript). The results of the EPR were further validated by comparing the XAFS spectroscopy data of both *D*-Cu-Crystal and *D*-Cu-SMOH samples (Figs. 2d-h in the revised manuscript). The information probed has not only provided in-depth insight into the disassembly evolution but also offered a well-defined framework platform for further study of structure-performance correlation as well as understanding the underlying mechanism.

Fig. R2. VCD and correlated normal IR spectra of *D(L)*-Cu-SMOHs. The purple shaded region represents signals from carboxylate group while the blue shaded region represents signals from Py ring.

(**Note:** This VCD has also been upgraded in the revised manuscript as Fig. 1m and corresponding IR spectra have been added to the revised *Supplementary Information* as Fig. S21.)

We have further employed vibrational circular dichroism (VCD) spectroscopy to provide direct evidence of retained helicity of *D(L)*-Cu-SMOHs. Distinct from electron transition-based ECD, VCD is ideally suited for discerning the intrinsic chirality of molecular conformations (i.e. helicity) thanks to its specific response to vibrations of molecular functional groups. The disassembled *D*-Cu-SMOH and

L-Cu-SMOH exhibit perfect mirror-image VCD spectra, robustly confirming their enantiomeric relationship (Fig. R2). Most notably, we observed strong VCD signals corresponding to skeletal vibrations of the intrinsically achiral pyridine ring (Py), especially 1620 (ν_2), 1595 (ν_3), 1562 (ν_4), 1392 (ν_6) and 1220 (ν_8) cm^{-1} (Table R2). The emergence of chiroptical activity from the achiral planar Py serves as unambiguous proof of its involvement in helical architecture.

In short, this integrated EPR, XAFS, and VCD characterization establishes a deeply insightful structural platform with well-defined compositions and structures, enabling further functional exploration and mechanism elucidation.

Table R2. Functional groups assignments for *D(L)*-SMOHs.

Band	IR (cm^{-1})	Assignment
ν_1	1641	C=O antisymmetric stretching of COO^-
ν_2	1620	C=N stretching of Py ring
ν_3	1595	C=C stretching of Py ring
ν_4	1562	C=C stretching of Py ring
ν_5	1430	C=O symmetric stretching of COO^-
ν_6	1392	C=C of Py ring
ν_7	1317	C–O stretching of COO^-
ν_8	1220	C–H stretching of Py ring

(Note: Table R2 has also been added to the revised *Supplementary Information* as Table S14.)

(3) This work systematically assesses the enantioselective HI inhibition performance of disassembled *D(L)*-Cu-SMOHs.

The unique disassembled state of *D(L)*-Cu-SMOHs provides enhanced exposure of chiral active sites and greater conformational adaptability than related bulk assemblies, enabling their further exploration for stereospecific amyloid fibrillization inhibition. The raw manuscript had undertaken a set of assays and characterizations of anti-amyloid efficacy of *D(L)*-Cu-SMOHs (Fig. 3 in the revised manuscript). To rigorously establish the anti-amyloid activity as an emergent property of *D(L)*-Cu-SMOHs rather than their isolated constituent parts, we have performed comprehensive controls using free *D(L)*-Py-Thr chiral ligands and another type of Cu-based framework

(Cu-Gly-MOF).

Critically, free ligands alone show negligible inhibitory activity, while Cu-Gly-MOF without aromatic group and chirality shows significantly reduced efficacy against HI fibrillization (Fig. R3, results summarized below). These control findings definitively establish that neither isolated chiral building blocks nor an achiral copper framework reproduces the significant and enantioselective HI fibrillization inhibition. As a result, this anti-amyloid function represents an emergent property arising exclusively from the well-defined chiral helical architecture of disassembled *D(L)*-Cu-SMOHs. This constitutes a novel functional paradigm for the reported disassembled fibrils, which was entirely absent in Ref. 14.

Fig. R3. Comprehensive controls of HI fibrillization in the presence of *D(L)*-Py-Thr ligands or Cu-Gly-MOF.

(Note: Fig. R3a, Figs. R3b-c, Figs. R3d-f, and Figs. R3g-i have also been added to the revised *Supplementary Information* as Fig. S35, Fig. S36, Fig. S37, and Fig. S38, respectively.)

(4) This work in-depth elucidates the chiral structure-performance relationship based on thermodynamic results and theoretical simulation.

Capitalizing on well-defined structures, another advancement in this work is elucidating correlations between unique chiral architectures of *D(L)*-Cu-SMOHs and their enantioselective anti-amyloid responses. Thermodynamic experiments (ITC and FL, Fig. 4) together with molecular docking simulations (Fig. 5) demonstrate that both *D*-Cu-SMOH and *L*-Cu-SMOH interact with HI via an unusual entropy-driven interaction ($\Delta H > 0$, $\Delta S \gg 0$), a characteristic signature of dominant hydrophobic forces (e.g., π - π stacking).

Fig. R4. ITC thermogram of titration of *D/L*-Py-Thr ligands, achiral Py-Gly ligand, or Cu-Gly-MOF (achiral and non-aromatic) into HI solution at 298.15 K. Fitting curves are obtained using the unit-point binding model.

(Note: Figs. R4a-d have also been added to the revised *Supplementary Information* as Figs. S41-S44, respectively.)

Table R3. Summarized ITC thermodynamics titrated at 298.15 K.

Sample	ΔH	ΔS	$T\Delta S$	ΔG
	$\text{kJ}\cdot\text{mol}^{-1}$	$\text{J}\cdot\text{mol}^{-1}\cdot\text{K}^{-1}$	$\text{kJ}\cdot\text{mol}^{-1}$	$\text{kJ}\cdot\text{mol}^{-1}$
HI+ D -Cu-SMOH	5.56	118.20	35.24	-29.68
HI+ L -Cu-SMOH	7.21	113.57	33.86	-26.65
HI+ D -Py-Thr	-7.29	62.74	18.71	-26.00
HI+ L -Py-Thr	-8.61	52.47	15.64	-24.25
HI+Py-Gly	-5.44	72.18	21.52	-26.96
HI+Cu-Gly-MOF	-35.77	-42.37	-12.63	-23.14

(Note: Table R3 has also been added to the revised *Supplementary Information* as Table S18.)

In the revised manuscript, supplemental controls (Fig. R4, Table R3) have been added

to provide definitive validation of the proposed interaction mechanism. In Fig. R4, free aromatic ligands, including *D/L*-Py-Thr and achiral Py-Gly, reveal fundamentally distinct exothermic binding ($\Delta H < 0$), indicative of electrostatic/hydrogen-bonding contributions. Crucially, their moderate entropy gains ($\Delta S > 0$) confirm preserved hydrophobic interactions from π - π stacking between Py rings and HI's hydrophobic domains. The most conclusive evidence comes from the achiral and non-aromatic Cu-Gly-MOF control, which displays totally reversed enthalpy-driven binding ($\Delta H < 0$, $\Delta S < 0$) accompanied by markedly reduced anti-amyloid efficacy. Collectively, the unique thermodynamic signature of *D(L)*-Cu-SMOHs ($\Delta H > 0$, $\Delta S \gg 0$) not only confirms a dominant hydrophobic π - π stacking mechanism but also establishes its critical role in inhibiting HI fibrillization.

In summary, this work advances beyond constructing hierarchical chiral framework nanostructures via a dynamic and reversible assembly-disassembly protocol. More critically, leveraging their well-defined structures, adaptive helical conformations, and thermodynamic interaction mechanisms, the disassembled single-unit chiral nanomaterials are anticipated to extend potent and chirality-dependent functionalities and performances unavailable from assembled counterparts.

Revision: The related experiment results have been added to the revised *Supplementary Information*. In order to further highlight the novelty and advancement of this work over previous ones, we made the following revisions in the main context of the revised manuscript (yellow-highlighted words).

On Page 4:

“Subsequent disassembly of crystalline *D(L)*-Cu-Crystals generates corresponding single-unit chiral metal-organic helices [*D(L)*-Cu-SMOHs]. It is worth emphasizing that conventional X-ray diffraction (XRD) and electron diffraction techniques face fundamental limitations in resolving ultrathin and aperiodic structures like disassembled single-unit fibrils^{14,19}. Herein, we introduce an integrated electron paramagnetic resonance (EPR) and X-ray absorption fine structure (XAFS)

methodology to unambiguously decipher structural parameters of *D(L)*-Cu-SMOHs, including metal coordination environment, electronic configuration, etc. Combined with helicity verification through vibrational circular dichroism (VCD), the well-defined *D(L)*-Cu-SMOHs hence establish a novel framework platform for further functional exploration as well as structure-performance correlation studies.”

On Page 5:

“To gain in-depth mechanistic insights, nanowatt isothermal titration calorimetry (ITC) and temperature-dependent fluorescence spectroscopy (FL) were employed to study the correlated thermodynamic interactions between amyloid and *D(L)*-Cu-SMOHs. The results revealed an unusual entropy-favored interaction mechanism driven by enantioselective hydrophobic π - π stacking, with *D*-Cu-SMOH exhibiting a more favored Gibbs energy change compared to its *L*-enantiomer. Furthermore, molecule docking simulation has further exquisitely depicted the significantly enantioselective π - π stacking between the aromatic pyridine group of *D(L)*-Cu-SMOHs and hydrophobic amino acid residues of amyloid those are of essential amyloidogenesis regions. In words, this work advances beyond constructing hierarchical chiral framework nanostructures via a dynamic and reversible assembly-disassembly protocol. More critically, leveraging their well-defined structures, adaptive helical conformations, and thermodynamic interaction mechanisms, the disassembled single-unit chiral nanomaterials are anticipated to extend potent and chirality-dependent functionalities and performances unavailable from assembled counterparts.”

Reviewer #2

General comments: In this work from Guo and co-workers, the authors report enantiopure helical copper -based coordination polymers and their effect on amyloid fibrillization. To keep it short, I find the reported work very interesting for a broad audience and well characterized, certainly worth publishing in Nature Communications. The results are supported by several well performed complementary analyses, each well described in a scholar manner, with an abundance of explanations and details.

Response: We sincerely thank the reviewer for his/her encouraging comments and constructive suggestions which help us further improve the rigor and quality of our manuscript. In response to the specific points raised, we have made the following revisions and a detailed point-to-point response below.

Specific comments:

1) Is the assembly-disassembly of the chiral coordination polymers reversible? From how it is depicted in Scheme 1 it should be reversible, however from the results there is no evidence of this. It seems that upon breaking the hydrogen-bonded framework it's not possible to go back. The authors should verify this, for instance supported by PXRD, otherwise scheme 1 should be changed accordingly.

Response: We thank the Reviewer for raising this very interesting question regarding assembly-disassembly reversibility. The dynamic hydrogen-bonding interactions intrinsically enable a unique reassembly of chiral hydrogen-bonded frameworks distinct from conventional MOFs/COFs. To demonstrate this, we introduced an aprotic solvent such as acetonitrile (ACN) into aqueous solutions of disassembled *D(L)*-Cu-SMOHs. ACN triggers water desolvation from *D(L)*-Cu-SMOHs and therefore promote inter-chain hydrogen bond reformation. Accordantly, crystalline frameworks reassembled from single-unit helices upon addition of 50% v/v ACN, as confirmed by PXRD and SEM analyses of the collected precipitates.

Fig. R1. Structural and morphological characterizations of *D(L)*-Cu-Reassemblies. (Note: Fig. R1a and Figs. R1b-g have also been added to the revised *Supplementary Information* as Fig. S26 and Fig. S27, respectively.)

As shown in Fig. R1, PXRD patterns of the collected precipitates closely match the calculated diffraction pattern from SXRD data of *D(L)*-Cu-Crystals, with characteristic peaks indexed to (004), (110), and (104) facet planes. SEM visualization of reassembled samples confirms restored 3D rectangular morphologies with dimensions of ca. 9.8 μm in both length and width and ca. 2.5 μm in height. Taking together, these data demonstrate that simple addition of aprotic acetonitrile triggers reassembly of *D(L)*-Cu-SMOHs into parent *D(L)*-Cu-crystal structures, unequivocally establishing the reversibility of the assembly-disassembly-reassembly cycle between *D(L)*-Cu-crystals and *D(L)*-Cu-SMOHs (Scheme 1 of the manuscript).

Revision: To highlight the unique assembly-disassembly reversibility of reported chiral hydrogen-bonded frameworks, we have added additional discussion on Page 13

in the revised manuscript (yellow-highlighted words). The supplemented PXRD curves and SEM analyses of reassembled samples have also been added as Fig. S26 and Fig. S27, respectively, in the revised *Supplementary Information*. Accordant reassembly method has also been added in the Method part on page 24.

On Page 13:

“H₂O molecules. Considering the dynamic feature of hydrogen bonding, we further explored whether disassembling reversibility occurs in *D(L)*-Cu-SMOHs. Very interestingly, introducing aprotic solvent (e.g., acetonitrile) triggers the water desolvation from *D(L)*-Cu-SMOHs, promoting interchain hydrogen bonding and reformation of blue precipitates (Details are available in Methods). PXRD (Fig. S26) and SEM (Fig. S27) characterizations of the centrifugated precipitates confirm reassembled crystalline structures identical to parent *D(L)*-Cu-Crystals, unequivocally demonstrating reversible assembly-disassembly cycling between *D(L)*-Cu-crystals and *D(L)*-Cu-SMOHs (Scheme 1).”

On Page 24:

“Reassembly of *D(L)*-Cu-SMOHs

Reassembly of the *D(L)*-Cu-SMOHs framework was executed by simply introducing an aprotic solvent. Specifically, ACN was added dropwise as the aprotic solvent to the aqueous dispersion of *D(L)*-Cu-SMOHs until achieving a 1:1 v/v ratio of ACN: H₂O. After 30 min of static incubation, blue precipitates formed and were isolated by centrifugation (5000 rpm, 5 min) for subsequent characterization.”

2) The PXRD spectra are calculated from SCXRD and not simulated. No computational method is employed but are directly calculated from the structure factors obtained from SCXRD, please correct this inaccuracy.

Response: We appreciate the Reviewer for his/her rigorous attention to detail regarding the terminology. The reference PXRD pattern was indeed calculated from experimental SXR data rather than computational modeling. We have corrected the terminology consistently throughout the manuscript.

Revision: Please see the updated Fig. 1 in the revised manuscript and Fig. S26 and Fig. S34 in the revised *Supplementary Information*.

Fig. 1. Dynamic evolution of assembly and disassembly of *D(L)*-Cu-Crystals. (a) Asymmetric coordination mode of *D*-Cu-Crystal with red asterisks indicating chiral centers. (b) The 1D helical unit featuring a left-handed 4_3 screw with a pitch of 35.6 Å along the *c*-axis direction. (c) 2D lamella structure assembled by interchain hydrogen bonding. (d) 3D framework structure projected in the *a*-*b* plane. Red dashed lines in (c) and (d) indicate hydrogen bonds. (e) PXRD patterns of *D(L)*-Cu-Crystals alongside calculated patterns. (f) and (g) DLS characterization of *D(L)*-Cu-SMOHs dispersed in water and inset show related Tyndall effect. (h) CD spectra of *D(L)*-Cu-SMOHs dispersed in water. (i) Corresponding crystallographic dimensions of *D*-Cu-SMOH single-unit helix according to SXR. (j) HR-TEM image of *D*-Cu-SMOH dispersed in water. (k) Corresponding crystallographic dimensions of *L*-Cu-SMOH single-unit helix according to SXR. (l) HR-TEM image of *L*-Cu-SMOH dispersed in H₂O. (m) VCD spectra of *D(L)*-Cu-SMOHs. The purple shaded region represents signals from carboxylate group while the blue shaded region represents signals from Py ring.

3) The CD spectra of the D and L-Cu-SMOH are well reported, however why no absorption spectra are reported? These should be added, also to appreciate more the strong gabs arising in the metal d-d transitions area.

Response: We thank the Reviewer for this constructive suggestion. While our original submission included the g-factor ($g_{Abs} = \frac{CD}{32980 \times Abs}$) plot as a function of wavelength derived from both CD and normal absorption data, the raw UV-Vis absorption spectra themselves were not presented. According to the Reviewer's suggestion, we have now included the UV-Vis absorption spectra for both *D*-Cu-SMOH and *L*-Cu-SMOH as Fig. S24 in the revised *Supplementary Information*.

Fig. R5. UV-Vis absorbance spectra of *D*-Cu-SMOH and *L*-Cu-SMOH. (Note: Fig. R5 has also been added to the revised *Supplementary Information* as Fig. S24.)

D-Cu-SMOH and *L*-Cu-SMOH exhibit intense absorption bands below 400 nm, mainly attributed to ligand-to-metal charge transfer (LMCT) and intra-ligand ($\pi \rightarrow \pi^*$ and $n \rightarrow \pi^*$) transitions. A broad *d-d* electronic transition band (ca. 2.2 eV) of central Cu(II) is observed within the visible region, which is consistent with the calculated $d_{yz}/d_{xz} - d_{x^2-y^2}$ transition gap (2.3 eV) for EO Cu(II) based on EPR results (Fig. 2c).

Revision: We have supplemented UV-Vis spectra of *D*-Cu-SMOH and *L*-Cu-SMOH

in the *Supplementary Information* (Fig. S24) and also briefly discussed these spectral features in the revised manuscript (yellow-highlighted words).

On Page 12:

“involved by H₂O coordination during the disassembled process³⁵. Note that, the estimated $d_{yz}/d_{xz} - d_{x^2-y^2}$ transition gap (2.3 eV) of EO Cu(II) shows good agreement with the observed visible $d-d$ absorption band (2.2 eV) of *D*-Cu-SMOH in Fig. S24.”

4) In the main text at page 24 row 312 the authors write: “Apparently, the retained CD intensity by *D*-Cu-SMOH is also stronger than that of *L*-Cu-SMOH, further demonstrating the higher enantioselective inhibition effect of *D*-Cu-SMOH on HI fibrillization.” However, the CD spectra in Fig. S22 are identical, while the data reported in the main, as fig. 3d and consequent TEM and AFM shows the different enantioselectivity. Is Fig. S22 reporting the correct data?

Response: We appreciate the Reviewer for his/her meticulous reading. The control experiment in Fig. S30 (formerly Fig. S22) depicts CD spectra of HI immediately after mixing with *D*-Cu-SMOH or *L*-Cu-SMOH, prior to incubation. Crucially, there are no CD variations at this initial stage, excluding any possible CD influences arising from *D(L)*-Cu-SMOHs themselves at the employed concentration. In sharp contrast, Fig. 3d demonstrates obvious CD variations after incubation of HI with *D(L)*-Cu-SMOHs, claiming potent inhibition of HI fibrillization. Thus, Fig. S30 shows correct data and serves as an essential control to exclude intrinsic CD contributions from *D(L)*-Cu-SMOHs.

Revision: To eliminate potential ambiguity, we have implemented the following revisions in the revised manuscript (yellow-highlighted words).

On Page 16:

“structure evolution of HI during fibrillization⁵¹. The pristine HI monomer exhibits

negative CD peaks around 208 nm and 222 nm which are assigned to the two characteristic signs of α -helix (Fig. 3d)⁵². As a pre-incubation control, the unaltered CD spectra of HI (Fig. S30) acquired immediately after addition of *D(L)*-Cu-SMOHs exclude intrinsic CD influences arising from themselves at the employed concentration.”

Moreover, an additional note “prior to incubation” has been added in Fig. S30 and the accordant caption has also been amended.

Fig. S30. Control CD spectra of HI in the presence/absence of *D*-Cu-SMOH or *L*-Cu-SMOH, were recorded immediately after mixing and prior to incubation. These data confirm the absence of immediate inhibitor-induced changes to HI’s secondary structure.

5) In part describing the interaction with amyloid, no control experiments with the ligands (*L*)-Py-Thr are reported. Why?

Response: We thank this constructive suggestion which is crucial for establishing the unique structure-dependent bioactivity of *D(L)*-Cu-SMOHs. Additional control experiments were performed using both *D*-Py-Thr and *L*-Py-Thr ligands (Figs. R6-R7). The inhibition performances have been comprehensively evaluated using ThT fluorescence assays (Fig. R6a), CD spectroscopy (Figs. R6b-c), TEM (Figs. R7a-b) and DLS (Figs. R7c-d). As results summarized below, free chiral ligands alone show negligible inhibitory activity.

Fig. R6. (a) Kinetics of HI Fibrillization monitored by the ThT assay in the absence/presence of *D(L)*-Py-Thr ligands. (b) CD spectra of HI with *D(L)*-Py-Thr ligands incubation and (c) secondary structure analysis from CD spectrum. (Note: Fig. R6a and Figs. R6b-c have also been added to the revised *Supplementary Information* as Fig. S35 and Fig. S36, respectively.)

ThT fluorescences for co-incubation of *D*-Py-Thr and *L*-Py-Thr with HI were nearly identical to the HI-only control (Fig. R6a), demonstrating their negligible inhibitory activity. CD spectroscopy and correlated secondary structure analysis further corroborated the negligible inhibition by control samples (Figs. R6b-c). HI incubated with *D*-Py-Thr or *L*-Py-Thr maintained characteristic β -sheet signatures analogous to fibrillated HI in Fig. 3d. Correlated β -sheet proportions ($\sim 38\%$) are found and not reduced compared to mature HI fibrils ($\sim 39\%$).

Fig. R7. TEM imaging of HI with *D*-Py-Thr ligand (a) or *L*-Py-Thr ligand (b) incubation. Size distribution of HI incubated in the presence of *D*-Py-Thr ligand (c) or *L*-Py-Thr ligand (d). (Note: Figs. R7a-b and Figs. R7c-d have also been added to the revised

Supplementary Information as Fig. S37 and Fig. S38, respectively.)

TEM micrographs of HI incubated with *D*-Py-Thr or *L*-Py-Thr also show mature amyloid fibrils (Figs. R7a-b). Corresponding DLS measurements confirm hydrodynamic dimensions of HI samples incubated with *D*-Py-Thr and *L*-Py-Thr (Figs. R7c-d) are also comparable to mature HI fibril (Fig. S33).

These collective results unequivocally demonstrate that neither free *D*-Py-Thr ligand nor free *L*-Py-Thr ligand exhibit obvious anti-amyloid activity. Therefore, the significant inhibitory effects observed for *D*-Cu-SMOH and *L*-Cu-SMOH can be confidently attributed to their assembled helical structures.

Revision: The control experiment results have been added in the *Supplementary Information*, please see Figs. S35-S38. More discussion has been added in the revised manuscript (yellow-highlighted words).

On Page 17:

“observed particularly in the case of *D*-Cu-SMOH. Control experiments using free chiral ligands (*D(L)*-Py-Thr) and an achiral Cu-based framework (Cu-Gly-MOF, Fig. S34)⁵⁹ further highlight the intrinsic anti-amyloid activity of *D(L)*-Cu-SMOHs. Complementary characterizations involving ThT assay (Fig. S35), CD spectroscopy (Fig. S36), TEM morphology (Fig. S37) and DLS analysis (Fig. S38) consistently show negligible or significantly reduced inhibitions of HI fibrillization by controls. These convergent results unambiguously establish the prominent enantioselective inhibition exhibited by *D(L)*-Cu-SMOHs.”

Reviewer #3

Comments:

General comments: The manuscript by Tang and co-workers introduces a biomimetic “assembly–disassembly” route to fabricate enantiopure, single unit Cu–organic helices whose chirality is preserved upon water triggered disassembly. The authors show that D and L helices inhibit insulin amyloid formation in vitro. The concept is interesting, but some control experiments will be useful to solidify the central findings of the work.

Response: We are very grateful to this reviewer for his/her positive assessment and insightful suggestions. We have considered these comments very seriously and have undertaken abundant experiments according to the suggestions. Please see the point-to-point response listed below.

Specific comments:

1) Entropy gains can also arise from release of structured water, counter ion effects or ligand reorganization. Thereby, the claim that $\Delta H > 0$ and $\Delta S > 0$ unambiguously prove a hydrophobic π – π stacking mechanism can be strengthened by more experiments. Entropy gains can also arise from release of structured water, counter ion effects or ligand reorganization. Some controls experiments (i.e. ITC of free *D(L)*-Py-Thr, achiral Cu-framework built with a non-aromatic ligand, a pyridyl ligand lacking chirality) will help to understand these contributions.

Response: We sincerely thank the reviewer for this insightful critique regarding in-depth thermodynamic interpretation. As mentioned by the reviewer, entropy increases in most cases stem primarily from release of structured water (i.e. desolvation effect) also well-known as the theory proposed by Ross and Subramanian (*Biochemistry* 1981, 20, 11, 3096–3102). In principle, hydrophobic interactions (like π – π stacking in this work) can be thought driven by the desolvation effect manifested as entropy gains from released water. To exclude other possible contributions, additional ITC controls have been conducted to unequivocally reveal the driving force between *D(L)*-Cu-SMOHs and HI.

(1) ITC of Free *D(L)*-Py-Thr Ligands with Blank Buffer (Negligible Heat Change):

Fig. R8. ITC thermograms in the titration of free *D*-Py-Thr ligand (a) and free *L*-Py-Thr ligand (b) into blank buffer at 298.15 K.

(Note: Fig. R8a and Fig. R8b have also been added to the revised *Supplementary Information* as Fig. S39 and Fig. S40, respectively.)

According to the suggestion, we first performed ITC titrations of free *D(L)*-Py-Thr ligands into blank buffer solution (Fig. R8). The experimental results showed negligible heat changes, excluding possible thermodynamic contributions arise from ligand self-organizations.

(2) ITC of free *D(L)*-Py-Thr ligands with HI ($\Delta H < 0$, $\Delta S > 0$):

Fig. R9. ITC thermograms in the titration of *D*-Py-Thr ligand (a) or *L*-Py-Thr ligand (b) into HI solution at 298.15 K. Fitting curves are obtained using the unit-point binding model.

(**Note:** Fig. R9a and Fig. R9b have also been added to the revised *Supplementary Information* as Fig. S41 and Fig. S42, respectively.)

We next conducted ITC titrations of free *D(L)*-Py-Thr ligands into HI (Fig. R9, Table R3). Distinct from the endothermic profiles present by *D(L)*-Cu-SMOHs ($\Delta H > 0$ and $\Delta S \gg 0$), free ligands present negative enthalpy changes ($\Delta H < 0$), suggesting exothermic interactions such as electrostatic interactions or/and hydrogen bonding. Though much smaller than that of *D(L)*-Cu-SMOHs (Table R3), positive entropy changes ($\Delta S > 0$) indicate hydrophobic interaction (i.e. π - π stacking) between pyridine ring of free *D(L)*-Py-Thr ligands and HI hydrophobic groups. This results hence strengthen the conclusion of π - π stacking occurring between similar pyridine moiety of *D(L)*-Cu-SMOHs with HI.

Table R3. Summarized ITC thermodynamics titrated at 298.15 K.

Sample	ΔH	ΔS	$T\Delta S$	ΔG
	$\text{kJ}\cdot\text{mol}^{-1}$	$\text{J}\cdot\text{mol}^{-1}\cdot\text{K}^{-1}$	$\text{kJ}\cdot\text{mol}^{-1}$	$\text{kJ}\cdot\text{mol}^{-1}$
HI+ D -Cu-SMOH	5.56	118.20	35.24	-29.68
HI+ L -Cu-SMOH	7.21	113.57	33.86	-26.65
HI+ D -Py-Thr	-7.29	62.74	18.71	-26.00
HI+ L -Py-Thr	-8.61	52.47	15.64	-24.25
HI+Py-Gly	-5.44	72.18	21.52	-26.96
HI+Cu-Gly-MOF	-35.77	-42.37	-12.63	-23.14

(**Note:** Table R3 has also been added to the revised *Supplementary Information* as Table S18.)

(3) ITC of Achiral Pyridyl Ligand (Py-Gly) with HI ($\Delta H < 0$, $\Delta S > 0$):

To isolate the hydrophobic Py contribution independent of chiral groups, we have synthesized additional achiral Py-Gly ligand by substituting glycine (achiral amino acid) for chiral threonine (Figs. S45-S48, and related method on Page 7 in the revised *Supplementary Information*). Similar to *D(L)*-Py-Thr, ITC titration of Py-Gly into HI (Fig. R10, Table R3) also reveals $\Delta H < 0$ and $\Delta S > 0$ thermogram. Critically, the positive ΔS term reinforced the entropy contribution from hydrophobic Py, though

diminished relative to *D(L)*-Cu-SMOHs.

Fig. R10. ITC thermograms in the titration of Py-Gly ligand into HI solution at 298.15 K. Fitting curves are obtained using the unit-point binding model. (Note: Fig. R10 has also been added to the revised *Supplementary Information* as Fig. S43.)

(4) ITC of Achiral and Non-Aromatic Cu-Framework (Cu-Gly-MOF) with HI ($\Delta H < 0$, $\Delta S < 0$):

Fig. R11. ITC thermograms in the titration of Cu-Gly-MOF into HI solution at 298.15 K. Fitting curves are obtained using the unit-point binding model.

(Note: Fig. R11 has also been added to the revised *Supplementary Information* as Fig. S44.)

We further sophiscatelly selected and synthesized a Cu-Gly-MOF free of both chirality and hydrophobic Py directly using achiral glycine as the ligand and copper (II) as the metal ion (Fig. S34). In stark contrast to the $\Delta H > 0$ and $\Delta S \gg 0$ profile presented by *D(L)*-Cu-SMOHs, ITC titration of Cu-Gly-MOF (Fig. R11, Table R3) into HI shows negative enthalpy change ($\Delta H < 0$) and strikingly negative entropy change ($\Delta S < 0$). This thermodynamic signature is classically recognized as strong enthalpically driving forces like electrostatic forcing or/and hydrogen bonding. The entropy inversion ($\Delta S < 0$) explicitly highlight the Py-determined hydrophobic interaction. Consistently, ThT assay (Fig. R12a), CD (Figs. R12b-c), TEM (Fig. R13c) and DLS (Fig. R13f) demonstrate substantially weaker inhibition by Cu-Gly-MOF than *D(L)*-Cu-SMOHs. Thus, absence of the Py group abolishes π - π stacking (yielding $\Delta S < 0$) and impairs fibrillization inhibition.

Based on a set of control experiments showed in above, the unique thermogram of *D(L)*-Cu-SMOHs ($\Delta H > 0$, $\Delta S \gg 0$) not only unequivocally claim a hydrophobic π - π stacking mechanism but crucially delineates its role in inhibiting HI fibrillization.

Revision: The revised version on Page 18 adds a description of the ITC control experiment and the following sentence:

“Totally distinct from the enthalpically favored interactions (e.g., electrostatic force or/and hydrogen bonding)⁶², our ITC results ($\Delta H > 0$ and $\Delta S \gg 0$) suggest an unusual entropy driven interaction (i.e. hydrophobic affinity stemming from desolvation effect) between HI and disassembled framework according to Ross and Subramanian's theory⁶³. To eliminate alternative entropy gains, a series of control ITC experiments (Figs. S39-S44) involving sole chiral *D(L)*-Py-Thr ligands ($\Delta H < 0$ and $\Delta S > 0$), achiral pyridyl-containing ligand (Py -Gly, Figs. S45-S48) ($\Delta H < 0$ and $\Delta S > 0$) and additional Cu-framework (Cu-Gly-MOF, Fig. S34) free of both chirality and aromaticity ($\Delta H < 0$ and $\Delta S < 0$) have been further conducted. According to results summarized in Table S18, all controls accordantly present exothermic thermograms

($\Delta H < 0$) in stark contrast to the unique endothermic signature ($\Delta H > 0$) of *D(L)*-Cu-SMOHs, thereby eliminating enthalpically driven interactions (e.g., electrostatic interactions and hydrogen bonding) between *D(L)*-Cu-SMOHs and HI. Furthermore, entropy gains ($\Delta S > 0$) persist in Py-containing ligands including *D(L)*-Py-Thr and Py-Gly while a reversed entropy decrease ($\Delta S < 0$) is observed for Cu-Gly-MOF due to without Py group, highlighting the Py-mediated hydrophobic interactions as the critical entropy source.”

2) The paper will benefit of a demonstration, through control experiments, that the chiral pyridyl threonine ligand or an achiral Cu framework alone do not exert similar anti amyloid effects.

Response: We thank the Reviewer for this constructive suggestion, which is crucial for establishing the unique structure-dependent inhibition by *D(L)*-Cu-SMOHs. We have added additional control experiments including assessments of *D(L)*-Py-Thr, achiral Cu-Gly-MOF (Fig. S34) in HI fibrillization. Comprehensive evaluation using ThT fluorescence assay (Fig. R12a), CD spectroscopy (Figs. R12b-c), TEM (Figs. R13a-c) and DLS (Figs. R13d-f) demonstrate negligible inhibitory activity for free *D/L*-Py-Thr ligands and significantly reduced inhibition for Cu-Gly-MOF.

Fig. R12. (a) Kinetics of HI Fibrillization monitored by the ThT assay in the absence/presence of *D(L)*-Py-Thr ligands or Cu-Gly-MOF. (b) CD spectra of HI with *D(L)*-Py-Thr ligands or Cu-Gly-MOF incubation and (c) secondary structure analysis from CD spectrum.

(Note: Fig. R12a and Figs. R12b-c have also been added to the revised **Supplementary Information** as Fig. S35 and Fig. S36, respectively.)

ThT fluorescences for co-incubation of *D*-Py-Thr and *L*-Py-Thr with HI were nearly

identical to the HI-only control (Fig. R12a), demonstrating their negligible inhibitory activity. For the achiral Cu-MOF sample, the final fluorescence intensity remained at 75% of the control level, contrasting sharply with the substantial inhibition ratio observed by *D*-Cu-SMOH (~30%) and *L*-Cu-SMOH (~40%). CD spectroscopy and correlated secondary structure analysis further corroborated the negligible inhibition by control samples (Fig. R12b-c). HI incubated with *D*-Py-Thr, *L*-Py-Thr, or achiral Cu-MOF maintained characteristic β -sheet signatures analogously to fibrillated HI in Fig. 3d. Correlated β -sheet proportions (~38%) are found and not reduced compared to mature HI fibrils (~39%).

Fig. R13. TEM imaging of HI with *D*-Py-Thr ligand (a), *L*-Py-Thr ligand (b), or Cu-Gly-MOF (c) incubation. Size distribution of HI incubated in the presence of *D*-Py-Thr ligand (d), *L*-Py-Thr ligand (e), or Cu-Gly-MOF (f).

(Note: Figs. R13a-c and Figs. R13d-f have also been added to the revised **Supplementary Information** as Fig. S37 and Fig. S38, respectively.)

TEM micrographs of HI incubated with *D*-Py-Thr, *L*-Py-Thr or achiral Cu-MOF also show mature amyloid fibrils (Figs. R13a-c). Corresponding DLS measurements confirm hydrodynamic dimensions of HI samples incubated with *D*-Py-Thr, *L*-Py-Thr and even achiral Cu-MOF (Figs. R13d-f) are also comparable to mature HI fibril (Fig. S33).

These collective results unequivocally demonstrate that neither free *D(L)*-Py-Thr

ligands nor achiral Cu-MOF exhibit obvious anti-amyloid activity. Therefore, the significant inhibitory effects observed for *D*-Cu-SMOH and *L*-Cu-SMOH can be confidently attributed to their assembled helical structures.

Revision: The control experiment results have been added in the *Supplementary Information*, please see Fig. S35-38. More discussion has been added in the revised manuscript (yellow-highlighted words).

On Page 17:

“observed particularly in the case of *D*-Cu-SMOH. Control experiments using free chiral ligands (*D(L)*-Py-Thr) and an achiral Cu-based framework (Cu-Gly-MOF, Fig. S34)⁵⁹ further highlight the intrinsic anti-amyloid activity of *D(L)*-Cu-SMOHs. Complementary characterizations involving ThT assay (Fig. S35), CD spectroscopy (Fig. S36), TEM morphology (Fig. S37) and DLS analysis (Fig. S38) consistently show negligible or significantly reduced inhibitions of HI fibrillization by controls. These convergent results unambiguously establish the prominent enantioselective inhibition exhibited by *D(L)*-Cu-SMOHs.”

3) Structural characterization (CD, DLS, TEM) should be also performed under physiological conditions.

Response: We thank the reviewer for this pertinent suggestion concerning the structural integrity of *D(L)*-Cu-SMOHs. We have performed structural characterizations (CD, DLS, and TEM) of both *D*-Cu-SMOH and *L*-Cu-SMOH under physiological conditions (10 mM phosphate-buffer, PB, pH=7.4, 37°C).

First of all, HR-TEM images (Figs. R14a-b) of *D*-Cu-SMOH and *L*-Cu-SMOH dispersed in PB at 37 °C reveal similar ultrathin fiber morphologies, with averaged diameters of 1.1 nm and averaged lengths of 25.2 nm, respectively. Note that minor noise spots in Figs. R14a-b may originate from residual buffer salts. Consistent nano-sized colloidal features are confirmed by DLS measurements (Figs. R14c-d), showing hydrodynamic sizes of 24.6 nm for *D*-Cu-SMOH and 26.1 nm for *L*-Cu-SMOH.

Fig. R14. HR-TEM image of *D*-Cu-SMOH (a) and *L*-Cu-SMOH (b) dispersed under physiological conditions. DLS characterization of *D*-Cu-SMOH (c) and *L*-Cu-SMOH (d) dispersed under physiological conditions (10 mM phosphate-buffer, PB, pH=7.4). (Note: Figs. R14a-b and Figs. R14c-d have also been added to the revised *Supplementary Information* as Fig. S17 and Fig. S18, respectively.)

Fig. R15. (a) CD spectra of *D(L)*-Cu-SMOHs dispersed under physiological conditions and (b) corresponding UV-Vis absorbance spectra.

(Note: Figs. R15 has also been added to the revised *Supplementary Information* as Fig. S19.)

Furthermore, CD spectra in PB (Fig. R15) closely match those in pure water (Fig. 3d), confirming preserved chirality. Collectively, these data clearly demonstrated well-maintained single-unit morphologies and chiral integrity for ***D(L)***-Cu-SMOHs under physiological conditions.

Revision: The structural characterizations (CD, DLS, TEM) of ***D(L)***-Cu-SMOHs performed under physiological conditions have been added in the revised ***Supplementary Information*** as the Figs. S17-S19. Corresponding discussions about structure stability have also been updated on Page 10 in the revised manuscript (yellow-highlighted words).

“Similar structural features (Figs. S17-S19) are also preserved for ***D(L)***-Cu-SMOHs under simulated physiological conditions (i.e. 10 mM phosphate buffer, pH = 7.4 and 37°C).”

4) The work will benefit of the quantification through vibrational circular dichroism or Raman optical activity analysis of the helicity and chiral density of SMOHs, and their correlation to inhibitory potency.

Response: We thank the reviewer for his/her insightful suggestion to characterize the helicity of ***D(L)***-Cu-SMOHs as well as to study their enantioselective inhibitory potency through vibrational circular dichroism (VCD). In comparison to electron transition-based ECD, VCD is widely accepted to be much less sensitive to local chirality. In return, acquired VCD signals together with assignable functional groups can be utilized as a reliable tool for probing intrinsic chiral conformations (e.g. helicity). (*J. Am. Chem. Soc.* 2012, 134, 26, 10974–10986)

VCD spectra of ***D***-Cu-SMOH and ***L***-Cu-SMOH (Fig. R3) exhibited mirror symmetric signatures, further solidifying their enantiomeric nature. After carefully assigning main vibration peaks according to normal IR (Fig. R3 and Table R2), ***D***-Cu-SMOH and ***L***-Cu-SMOH both exhibit pronounced split-type Cotton signals (purple shaded region) at 1641 (ν_1), 1430 (ν_5), and 1317 (ν_7) cm^{-1} , attributed to the C=O and C–O stretching vibrations originating from the carboxylate (COO^-) group. Additional

non-split VCD signals (blue shaded region) around 1620 (ν_2), 1595 (ν_3), 1562 (ν_4), 1392 (ν_6) and 1220 (ν_8) are also clearly observed, which are assigned to skeleton vibrations of the intuitively achiral pyridine ring. Note that such prominent VCD signatures of planar pyridine ring hence claim intrinsic helical conformations in *D(L)*-Cu-SMOHs.

Fig. R2. VCD and correlated normal IR spectra of *D(L)*-Cu-SMOHs. The purple shaded region represents signals from carboxylate group while the blue shaded region represents signals from Py ring.

(Note: This VCD has also been upgraded in the revised manuscript as Fig. 1m and corresponding IR spectra have been added to the revised *Supplementary Information* as Fig. S21.)

Table R2. Functional groups assignments for *D(L)*-SMOHs.

Band	IR (cm ⁻¹)	Assignment
ν_1	1641	C=O antisymmetric stretching of COO ⁻
ν_2	1620	C=N stretching of Py ring
ν_3	1595	C=C stretching of Py ring
ν_4	1562	C=C stretching of Py ring
ν_5	1430	C=O symmetric stretching of COO ⁻
ν_6	1392	C=C of Py ring
ν_7	1317	C–O stretching of COO ⁻
ν_8	1220	C–H stretching of Py ring

(Note: Table R2 have also been added to the revised *Supplementary Information* as

Table S14.)

Furthermore, VCD spectroscopy (Fig. R16) demonstrates enantioselective inhibition by *D(L)*-Cu-SMOHs more effectively than conventional IR. The VCD spectra of HI monomer shows α -helical characteristics with VCD bands appearing at 1650(+) and 1635(-) cm^{-1} , which obviously shift to distinct definitive β -sheet signatures at 1677(+), 1660(+), 1648(-), and 1621(+) cm^{-1} upon fibrillization. Crucially, HI maintains its α -helical VCD profile with *D*-Cu-SMOH, confirming potent fibrillization inhibition. In contrast, *L*-Cu-SMOH permits partial HI fibrillization, evidenced by intermediate β -sheet signals at 1665(+) and 1648(-) cm^{-1} . These distinct VCD responses reveal superior efficacy of *D*-Cu-SMOH over its *L*-enantiomer, consistent with other assays and confirming enantioselective inhibition.

Fig. R16. VCD spectra of HI without/with *D(L)*-Cu-SMOHs incubation.

(Note: Figs. R16 has also been added to the revised *Supplementary Information* as Fig. S31.)

Revision: The revised manuscript now includes a description of the VCD spectra of *D(L)*-Cu-SMOHs, as well as a description of the VCD experimental results pertaining to HI fibrillation:

The description of the VCD spectra of *D(L)*-Cu-SMOHs on Page 10 in the revised manuscript (yellow-highlighted words):

“We further employed vibrational circular dichroism (VCD) spectroscopy to resolve whether helical conformation is preserved by *D(L)*-Cu-SMOHs^{30,31}. Remarkably,

mirror-symmetric VCD signals (Fig. 1m), especially those corresponding to skeleton vibrations of the planar and axial pyridine ring (Fig. S21 and Table S14), definitively claim its involvement in intrinsic helical conformations.”

The description of the VCD experimental results pertaining to HI fibrillation on Page 17 in the revised manuscript (yellow-highlighted words):

There are no obvious vibration peaks around 1630 cm^{-1} under introductions of *D(L)*-Cu-SMOHs particularly for *D*-Cu-SMOH, reaffirming a stronger inhibition effect. “Consistent enantioselective inhibitory potency can be more remarkably discerned using VCD spectroscopy (Fig. S31).”

Response to Reviewers' Comments

Reviewer #1

General comments:

I appreciate the authors' responses and their efforts to improve the quality of the manuscript and to provide additional experimental proofs to support their hypotheses and conclusions. I am still not totally convinced that this work should be published in Nat. Commun. since a very similar work by the same authors (ref 14) has been published in Nat. Commun. in 2024, but let's say this revised manuscript could be published.

Response: We thank the reviewer for his/her invaluable approval to our revised manuscript and kind recommendation.

Reviewer #2

General comments:

In this revised submission of the work from Guo and co-workers, the authors answered all my questions in a satisfactory manner. I believe that the extra experiment added improved the manuscript further, and my previous revision was already positive. The assembly/disassembly process is proven in a smart way, using acetonitrile as a solvent and well supported by PXRD and SEM data. The updated discussion on the absorption data matches well the EPR data. My concern about CD experiments has been clarified in both the main text and supporting information. Finally, the control experiments with only the ligand add a solid extra experiment to the overall findings.

Despite the similarity with the author's previous work with Zn(II), in this manuscript the findings and characterization is interesting enough and fits the scope of the journal. Therefore, I believe this revised version is worth publishing as it is in Nature Communications.

Response: We are very grateful for the reviewer's positive evaluation and kind recommendation.

Reviewer #3

General comments:

The authors have addressed all the concerns raised in the previous round and the paper is now suitable for publication.

Response: We thank the reviewer for his/her positive recognition and kind recommendation to our revised manuscript.